# Direct production of olefins from syngas with ultrahigh carbon efficiency

Hailing Yu [1,2,6], Caiqi Wang [1,2,6], Tiejun Lin [1,6], Yunlei An[1], Yuchen Wang[1,3], Qingyu Chang[1], Fei Yu[1], Yao Wei[2,4], Fanfei Sun[5], Zheng Jiang[5], Shenggang Li [1,3], Yuhan Sun [1,3] ✉ & Liangshu Zhong [1,3] ✉

Syngas conversion serves as a competitive strategy to produce olefins chemicals from nonpetroleum resources. However, the goal to achieve desirable olefins selectivity with limited undesired C1 by-products remains a grand challenge. Herein, we present a non-classical Fischer-Tropsch to olefins process featuring high carbon efficiency that realizes 80.1% olefins selectivity with ultralow total selectivity of $CH_4$ and $CO_2$ (<5%) at CO conversion of 45.8%. This is enabled by sodium-promoted metallic ruthenium (Ru) nanoparticles with negligible water-gas-shift reactivity. Change in the local electronic structure and the decreased reactivity of chemisorbed H species on Ru surfaces tailor the reaction pathway to favor olefins production. No obvious deactivation is observed within 550 hours and the pellet catalyst also exhibits excellent catalytic performance in a pilot-scale reactor, suggesting promising practical applications.

Olefins including lower olefins ($C_{2-4}^=$) and long-chain olefins ($C_{5+}^=$, olefins with five or more carbon atoms) are important feedstocks in chemical industry for the production of plastics and basic chemicals[1]. Commercially, lower olefins are mainly produced by cracking of naphtha or pyrolysis of light alkanes, while oligomerization of lower olefins leads to high value-added long-chain olefins[2–4]. The limited petroleum resources and the growing market demand for olefins stir the development of alternative routes for olefins production from nonpetroleum feedstocks. Fischer-Tropsch to olefins (FTO) is a highly efficient technology to produce olefins directly from syngas − a mixture of hydrogen and carbon monoxide derived from coal, natural gas, biomass, solid waste, and $CO_2$ through commercially mature gasification/reforming technology[5]. However, the goal to achieve desirable olefins selectivity, especially for $C_{5+}^=$ slate, with limited undesired C1 by-products remains a grand challenge.

Recently, direct conversion of syngas to olefins (STO) has been well explored with significant progress. One of the highlighted routes is based on the bifunctional catalysis using oxide-zeolite composite catalyst (OX-ZEO)[6–8], where CO activation and C-C coupling are performed on separated active sites, enabling selectivity to lower olefins up to ~80% in hydrocarbons with CO conversion less than 20% under 673 K. Fischer-Tropsch to olefins (FTO) provides another direct route for olefins production from syngas[1,2,9–11], which is highly efficient for long-chain olefins production. The classic Fischer-Tropsch synthesis (FTS) process mainly produces heavy saturated hydrocarbons over a variety of metal catalysts including iron, cobalt, and ruthenium[12]. To date, only promoted iron- or cobalt carbide catalysts can effectively catalyze FTO reaction with selectivity to lower olefins in hydrocarbons up to 60%[9,13,14]. Nevertheless, both OX-ZEO and metal carbide-based FTO processes commonly exhibit high selectivity to $CO_2$ by-products in the range of 30%-50%, significantly decreasing the carbon utilization efficiency[7,11,15,16]. The olefins selectivity reported above would decline to below 60% when being calculated by considering the presence of $CO_2$ (Fig. 1a and Supplementary Table 1). In addition, from the viewpoint of practical application, the large amount of $CO_2$ produced during STO process not only decreases CO conversion, therefore needing a higher

[1]CAS Key Laboratory of Low-Carbon Conversion Science and Engineering, Shanghai Advanced Research Institute, Chinese Academy of Sciences, Shanghai 201210, P. R. China. [2]University of the Chinese Academy of Sciences, Beijing 100049, P. R. China. [3]School of Physical Science and Technology, ShanghaiTech University, Shanghai 201210, P. R. China. [4]Shanghai Institute of Applied Physics, Chinese Academy of Sciences, Shanghai 201800, P. R. China. [5]Shanghai Synchrotron Radiation Facility, Shanghai Advanced Research Institute, Chinese Academy of Sciences, Shanghai 201210, P. R. China. [6]These authors contributed equally: Hailing Yu, Caiqi Wang, Tiejun Lin. ✉e-mail: sunyh@sari.ac.cn; zhongls@sari.ac.cn

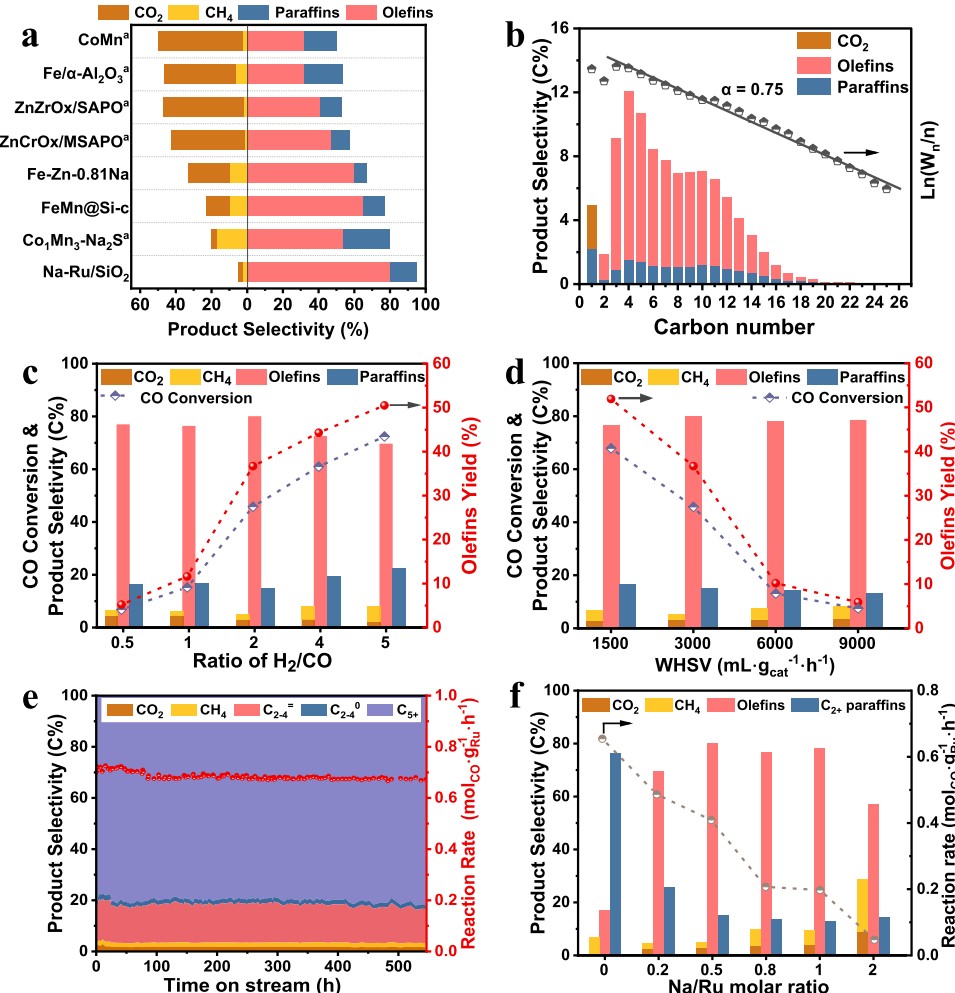

**Fig. 1 | Catalytic performance for direct syngas conversion to olefins.**
**a** Comparison of catalytic performance among Na-Ru/SiO$_2$ and other previously reported catalysts[7–11,13,15]. (a: C$_{2-4}^=$ selectivity). **b** Detailed product distribution (including CO$_2$) and ASF distribution of hydrocarbons over Na-Ru/SiO$_2$ catalyst. **c** Product selectivity, CO conversion and olefins yield at different H$_2$/CO ratios in syngas over Na-Ru/SiO$_2$ catalyst at 533 K, 3000 mL·g$_{cat.}^{-1}$·h$^{-1}$, and 1.0 MPa. **d** Product selectivity, CO conversion and olefins yield at different space velocities over Na-Ru/SiO$_2$ catalyst at 533 K, H$_2$/CO ratio of 2 and 1.0 MPa. **e** Stability test for Na-2%Ru(P)/SiO$_2$ catalyst. **f** Reaction rate of CO and product selectivity at different Na/Ru molar ratios. Reaction conditions: 533 K, 1.0 MPa, 3000 mL·g$_{cat.}^{-1}$·h$^{-1}$, H$_2$/CO ratio of 2.

recycle rate to obtain high overall CO conversion, but also requires decarburizing unit to separate the generated CO$_2$ in the recycling gas, leading to additional energy consumption in the whole plant. Many recent works try to address this challenge by tailoring the surface properties of catalysts or developing modified FTS catalysts featuring low intrinsic WGS reactivity[11,15–17]. For example, Xu et al. prepared a hydrophobic core-shell FeMn@Si-c catalyst, which can suppress the total selectivity of CO$_2$ and CH$_4$ to ~23% while remaining ~65% of olefins selectivity at CO conversion of 56.1%[11]. Xie et al. indicated that a Na/S/Mn modified hcp Co presented 54% of selectivity to lower olefins with 17% of CH$_4$ selectivity and <3% of CO$_2$ fraction at 1% CO conversion[15]. Despite these promising results, carbon efficiency is still low and the goal to reach the maximum olefins (especially for C$_{5+}^=$) selectivity and yield while simultaneously minimizing the production of undesired C1 by-products including CH$_4$ and CO$_2$ at a considerable activity level challenge the current FTO technology.

Herein, we present a silica-supported Ru nanoparticles (NPs) catalyst with sodium (Na) as the promoter (denoting as Na-Ru/SiO$_2$), which was highly active for FTO reaction but very inactive for WGS reaction. With various characterizations and surface probe reaction experiments, the Ru metal was demonstrated to be the active phase, and the Na promoter can suppress the reactivity of chemisorbed H atoms on Ru surface sites while greatly promote olefins production.

The results of this work demonstrate that the modified-metallic Ru can effectively tune the dominated product distribution from traditional paraffins to value-added olefins with sufficient selectivity and yields to justify the promising potential industrial applications.

## Results

### Catalytic performance
The catalytic performance of Na-Ru/SiO$_2$ catalyst with 5% of theoretic weight of Ru loading and 0.5 molar ratio of Na/Ru was evaluated at 533 K, 1.0 MPa and H$_2$/CO ratio of 2. An unexpected high olefins selectivity up to 80.1% was achieved at a CO conversion of 45.8% with selectivity to CO$_2$ and CH$_4$ limited within 2.7% and 2.2%, respectively over Na-Ru/SiO$_2$ catalyst. (Fig. 1a and Supplementary Table 1). The as-obtained CH$_4$ selectivity was far below the value predicted by the Anderson-Schulz-Flory (ASF) rule, and the chain-growth probability (α) for hydrocarbon products was as high as 0.75, indicating its suitability for long-chain olefins production, in good agreement with the observation of 74.5% of C$_{5+}^=$ slate in the olefins distribution (Fig. 1b). Specially, the fraction of value-added C$_5$-C$_{11}$ α-olefins was as high as 57.8%, which can be used for the production of high-quality lubricant, plasticizer and surfactant, while the fraction of detergent-range C$_{12}$-C$_{18}$ α-olefins reached 16.4% (Supplementary Fig. 1). This type of catalyst is also very suitable for converting methane-derived H$_2$-rich syngas to

long-chain olefins. A higher $H_2/CO$ ratio benefited CO conversion, which increased to 72.4% at $H_2/CO$ ratio of 5, for instance, whereas the olefins selectivity maintained above 70% without significant changes in the fraction of C1 by-products (Fig. 1c, Supplementary Table 2). The investigation of the effect of space velocity on catalytic performance suggested that the single-pass yield of olefins can reach up to 51.9% with 6.7% of C1 by-products selectivity at a CO conversion of 67.9% under 1500 mL·$g_{cat.}^{-1}$·$h^{-1}$·(Fig. 1d, Supplementary Table 3). The increase of reaction pressure, however, could decrease olefins selectivity from 81.0% at 0.5 MPa to 53.5% at 3 MPa (Supplementary Table 4). Furthermore, the stability test was carried out as shown in Supplementary Fig. 2. The catalytic performance for both $Ru/SiO_2$ and $Na-Ru/SiO_2$ catalysts remained stable within 50 h of test. Especially, the Na-2% Ru(P)/$SiO_2$ catalyst with much lower Ru loading amount (1.8 wt.% Ru, ICP) exhibited high stability for 500 h without any significant loss in activity and selectivity. Overall, the activity remained at around 0.700 $mol_{CO}$·$g_{Ru}^{-1}$·$h^{-1}$ with intrinsic TOF of 0.210 $s^{-1}$, and the olefins selectivity in total products kept in the range of 75-80% while that of undesired C1 by-products was always suppressed within 5% (Fig. 1e and Supplementary Fig. 3). Compared with the reported results for current STO catalysts under various CO conversion levels (Fig. 1a and Supplementary Table 1), the as-obtained $Na-Ru/SiO_2$ catalyst exhibits the highest olefins (especial for $C_{5+}^{=}$) selectivity and yield together with the lowest fraction of undesired C1 by-products including $CH_4$ and $CO_2$.

Noteworthy, the catalytic behavior of $Na-Ru/SiO_2$ is quite different from that of a conventional Ru-based FTS catalyst, which mainly produces saturated hydrocarbons instead of olefins[18–20]. As shown in Fig. 1f and Supplementary Fig. 4, the typical $Ru/SiO_2$ catalyst without Na promoter exhibits a very high CO conversion (73.3%) with 76.5% of paraffins selectivity and rather low selectivity to olefins (16.9%). After introducing Na promoter, the activity, chain-growth probability (α), and $CH_4$ selectivity decreased greatly, while olefins selectivity surprisingly increased to >70% for the sample with a Na/Ru molar ratio of more than 0.2, which reached a maximum value of 80.1% for the sample with Na/Ru molar ratio of 0.5. Furthermore, the product selectivity was compared at similar conversion levels, as shown in Supplementary Fig. 5. Under similar CO conversion of ~70%, the sample of $Na-Ru/SiO_2$ still exhibited high olefins selectivity of ~76% with suppressed C1 by-products, while a large amount of paraffins with selectivity of ~76% were produced over $Ru/SiO_2$ case. The detailed comparison of each C-containing hydrocarbon product in Fig. 1b and Supplementary Fig. 4 showed that more primary linear olefins than paraffins were produced for $Na-Ru/SiO_2$, suggesting that β-H elimination dominated the carbon chain termination pathway according to the Fischer-Tropsch reaction mechanism (Supplementary Fig. 6) and the secondary hydrogenation of olefins was also significantly inhibited. The catalytic performance at varied reaction temperatures further revealed the higher selectivity and yield for olefins over $Na-Ru/SiO_2$ catalyst than its $Ru/SiO_2$ counterpart (Supplementary Fig. 7, Supplementary Table 5). However, the excessive Na concentration, i.e., Na/Ru atomic ratio of 2, would decline the olefins selectivity significantly and simultaneously cause C1 by-products fraction to surge to ~30%. Obviously, suitable amount of Na doping plays a vital role in tuning the reaction pathway to favor olefins production with inhibited formation rate of C1 by-products and saturated hydrocarbons. Similar promotion effect for olefins production with high selectivity was also observed for other alkali metal promoters (Li, K, Rb, Cs). The increase of atomic number from Li to Cs caused a decreased trend for CO conversion but inversely promoted the formation of long-chain olefins slate from 72.7% for $Li-Ru/SiO_2$ to 83.0% for $Cs-Ru/SiO_2$ in olefins distribution (Supplementary Table 6). Overall, the alkali-promoted $Ru/SiO_2$ catalysts cause the transformation of catalytic performance from typical FTS regime to non-classic FTO with ultrahigh olefins selectivity and limited C1 by-products, which is rarely reported in the previous studies[18,21,22].

## Structure characterizations

In order to reveal the nature of the active sites that favor the formation of olefins, we resorted to several characterization approaches to investigate the detailed catalyst structure. Ex situ X-ray diffraction (XRD) and in situ XRD results both suggested that $RuO_2$ in fresh $Na-Ru/SiO_2$ was completely transformed into metallic Ru phase after $H_2$ reduction at temperature >573 K, which remained unchanged during the FTO reaction process (Fig. 2a and Supplementary Fig. 8). High resolution transmission electron microscopy (HRTEM) images confirmed the existence of metallic Ru with interplanar distance of (101) at around 2.04 Å and the size of Ru NPs slightly increased from $4.7 \pm 1.0$ nm to $5.2 \pm 1.2$ nm after reaction (Fig. 2b, c). Similar metallic Ru phase was observed for spent $Ru/SiO_2$ with size of Ru NPs centering at around 8.4 nm (Supplementary Fig. 9). A qualitative comparison of the exposure of Ru species can be obtained by estimating the Ru dispersion through TEM and/or CO chemisorption (Supplementary Table 7). Compared with $Ru/SiO_2$, higher metal dispersion (i.e. 11.0%) and metallic surface area (i.e. 49.3 $m^2/g_{Ru}$) were obtained for $Na-Ru/SiO_2$. It was thus suggested that the addition of Na during catalyst preparation could effectively facilitate the dispersion of Ru NPs, in good agreement with XRD &TEM results and previous reports[23,24]. The evolution of the chemical state of $Na-Ru/SiO_2$ and $Ru/SiO_2$ catalysts during the reduction and FTO reaction process was investigated using in situ X-ray adsorption spectroscopy (XAS). Figure 2d presents the R-space of Fourier transform (FT) extended X-ray adsorption fine structure (EXAFS) spectra at different stages. For the fresh $Na-Ru/SiO_2$ sample at room temperature ($H_2$-298 K), a major peak corresponding to Ru-O coordination was observed at ~1.5 Å. As the temperature increased to >423 K under $H_2$ flow, the peak for Ru-O disappeared. However, the major peak attributed to Ru-Ru pair at ~2.4 Å was observed, closing to that of metallic Ru foil. The above results indicated the complete reduction of $RuO_2$ to Ru metal phase. After $H_2$-treatment at 573 K, the temperature was decreased to 533 K and the atmosphere was switched to syngas ($H_2/CO = 2$) to explore the chemical variation of active Ru centers during the FTO reaction. A very analogous FT EXAFS spectrum was obtained, suggesting an unchanged local geometric structure of the Ru species. The EXAFS fitting results further confirmed the existence of Ru metal phase after reduction and reaction, which is independent of Na doping (Supplementary Fig. 10 and Supplementary Table 8). The change of electronic structure was further analyzed via in situ X-ray absorption near-edge structure (XANES) in Fig. 2e and Supplementary Fig. 11. The absorption edge distinctly shifted toward a lower energy closing to Ru foil for Na-promoted and unpromoted samples, suggesting the reduction of oxidation state of Ru atoms. The similar XANES spectra of $Na-Ru/SiO_2$ after reduction and reaction revealed the unchanged metallic state of Ru species, agreeing well with the in situ EXAFS results.

Furthermore, we found that the introduction of Na greatly improved the dispersion of Ru NPs. Specially, the Na promoter was homogenously distributed on both $SiO_2$ support and Ru NPs. A higher density of Na promoter can also be clearly identified as Na loading amount increases (Supplementary Fig. 12). The homogeneous distribution of Na over the catalyst surface may benefit the strong electronic interaction between Ru NPs and Na. To confirm this point, we used X-ray photoelectron spectroscopy (XPS) to examine the electronic state of Ru center. As shown in Supplementary Fig. 13, $Ru^{4+}$ was observed for both fresh samples. After reduction, the $Ru^0$ $3d_{5/2}$ peak of $Na-Ru/SiO_2$ at 279.9 eV exhibited a 0.5 eV lower binding energy than that of $Ru/SiO_2$ sample (280.4 eV), implying the increased electronic density over Ru metal phase due to the electron transfer from Na promoter to near-surface $Ru^0$ atom (Fig. 2f). The change of Ru electronic state was further demonstrated by CO-DRIFTS spectra (Fig. 2g). The peak at 2053 $cm^{-1}$ was ascribed to linearly coordinated CO on $Ru^0$ center over $Ru/SiO_2$ sample. However, the peak shifted toward a lower wavenumber at 2039 $cm^{-1}$ for $Na-Ru/SiO_2$, reflecting an increase of π*

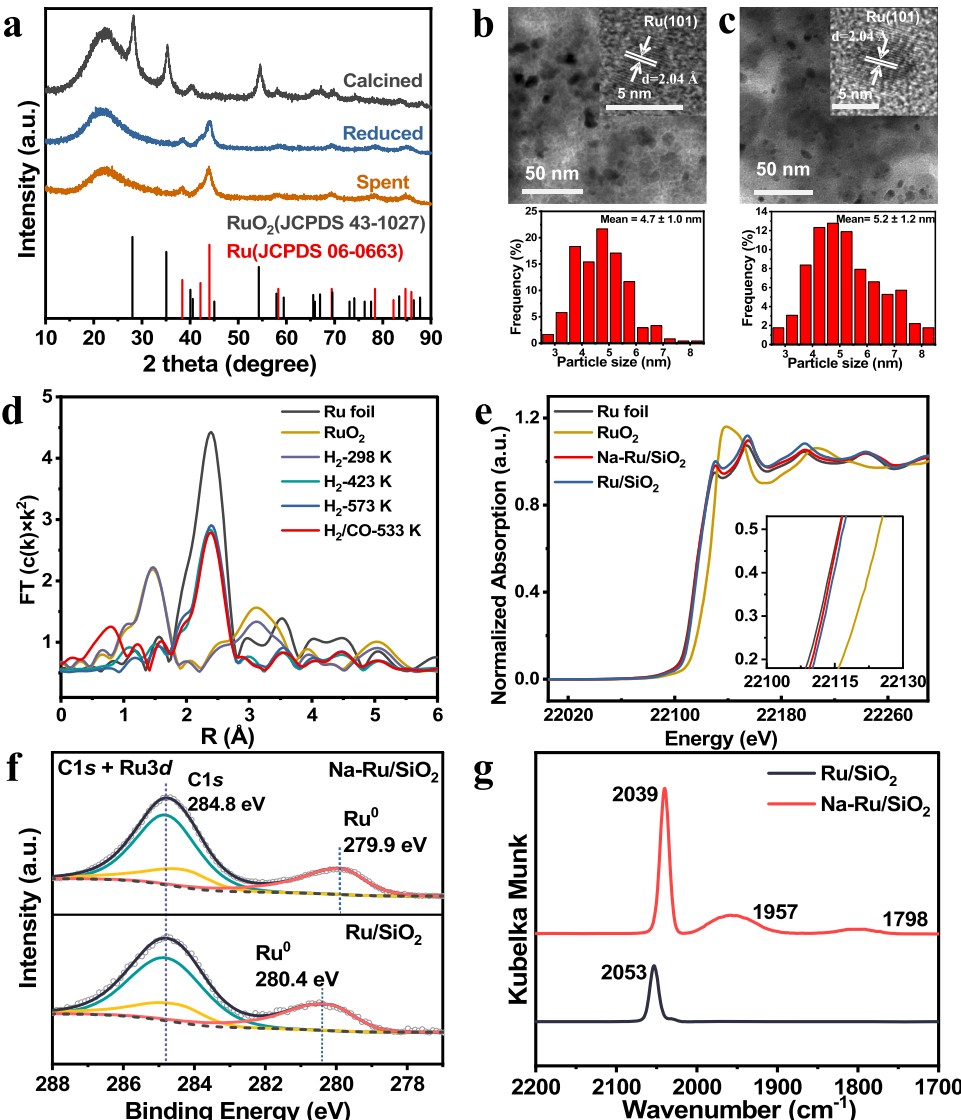

**Fig. 2 | Characterization of Na-Ru/SiO₂ and Ru/SiO₂ catalysts. a** XRD patterns of Na-Ru/SiO₂ at different stages. **b, c** (HR)TEM images and size distribution of Ru NPs for Na-Ru/SiO₂. **b** Reduced sample; **c** Spent sample. [Insets: Lattice fringes with distance of 2.04 Å corresponding to the Ru (101) crystal plane shown in (**b, c**)]. **d** Ru K-edge FT EXAFS spectra under different conditions on Na-Ru/SiO₂. **e** Ru K-edge X-ray absorption near-edge structure (XANES) spectra of Na-Ru/SiO₂, Ru/SiO₂, RuO₂, and Ru foil. **f** C 1s and Ru 3d photoemission spectra of reduced Ru/SiO₂ and Na-Ru/SiO₂. **g** DRIFTS spectra of adsorbed CO on Ru/SiO₂ and Na-Ru/SiO₂ at 323 K.

back-donation from Ru atoms to adsorbed CO*[25]. This result confirmed the formation of electron-rich Ru centers for Na-Ru/SiO₂ sample. Additionally, the peak at 1798 cm⁻¹ is ascribed to bridge-bound CO*[26], and the peak at 1957 cm⁻¹ is generally associated with CO adsorbed on Ru-support interface sites[27,28]. The much stronger peak intensity of linearly-bonded CO* and the appearance of another two adsorption peaks for Na-Ru/SiO₂ catalyst suggested that the presence of Na promoter strengthens the CO adsorption capacity, in line with the observed increased CO uptake for Na-Ru/SiO₂ than that of Ru/SiO₂ (Supplementary Table 7). Moreover, the Bader charge analysis based on DFT calculations showed that each Na₂O moiety donates a total charge of −0.64 |e| to the adjacent Ru atom when loading Na ion on the Ru (0001) surface, making the surface Ru metal species electron-rich (Supplementary Fig. 14).

## Structure-performance relationship

Based on the linear characteristic of ASF distribution and the higher chain-growth probability (α) as well as the ultralow CH₄ selectivity, it can be inferred that both Ru/SiO₂ and Na-Ru/SiO₂ might follow the analogous reaction mechanism[26,29,30]. This can be rationalized from the simplified surface carbide mechanism (Supplementary Fig. 6), which is widely accepted for the FTS[31]. Typically, the dissociated CO would be hydrogenated to form CHₓ as the main surface intermediate for chain propagation on the metallic Ru surface. The carbon chain grows by coupling of CHₓ units to the adsorbed alkyl-chain species. The chain growth is terminated by hydrogenation to produce paraffins or β-hydride abstraction to form olefins. We speculate that the possible reason for the huge difference in the catalytic performance of Ru/SiO₂ and Na-Ru/SiO₂, including activity and selectivity may lie in the discrepancy of dynamic of chemisorbed hydrogen[23], which alters the carbon-chain termination pathway. Specifically, the Na promoter significantly weakens the hydrogenation ability of Ru metal surface, thus β-H elimination dominates the carbon-chain termination pathway with the suppression of secondary hydrogenation of olefins. To verify our hypothesis, we designed several experiments to explore the reactivity and mobility of chemisorbed hydrogen. H₂-TPR results indicated that the presence of Na would suppress the reduction of RuO₂ to Ru metal (Supplementary Fig. 15), which can be attributed to the lower abundance of adsorbed H₂ or the reduced surface mobility of chemisorbed H atoms. The evolution of CO_ad species during H₂ flow at 533 K

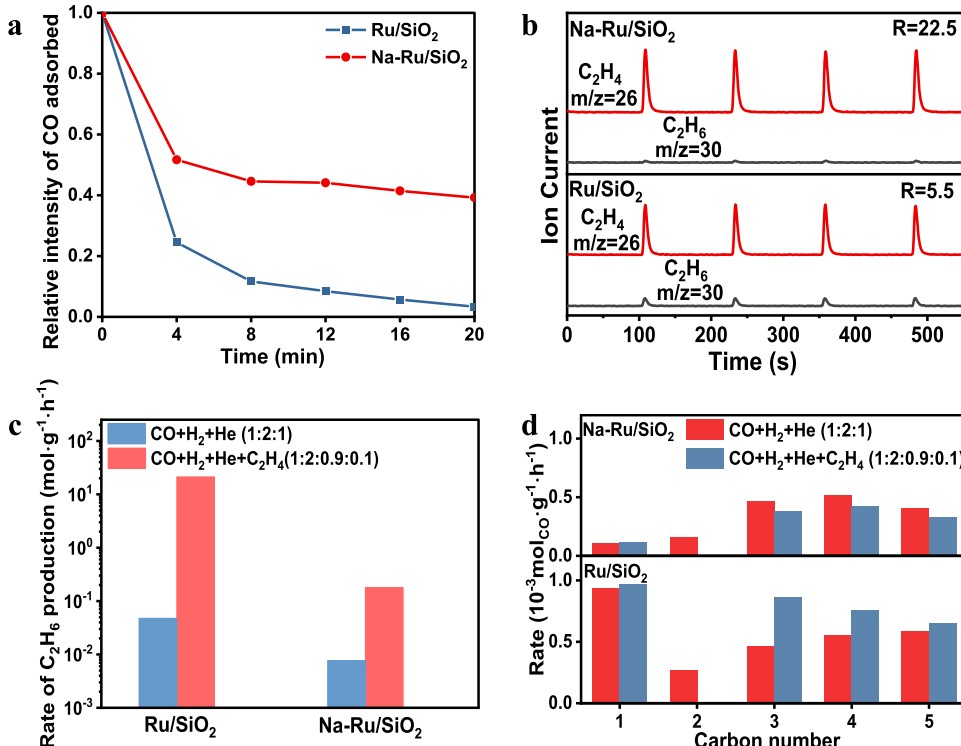

**Fig. 3 | Exploration of the reactivity of chemisorbed $H_2$ over Ru/SiO$_2$ and Na-Ru/SiO$_2$. a** Relative intensity of linear-adsorbed CO remained during stepwise hydrogenation at 533 K determined by in situ DRIFTS. The ordinates are the evolution of integrated peak area of $CO_{ad}$ species under $H_2$ flow divided by that before $H_2$ addition at 533 K. **b** Transient response curves obtained during pulses of 370 μL pure $C_2H_4$ into a flow of diluted $H_2$ (10% $H_2$, 90% Ar, 20 mL·min$^{-1}$) at 533 K. R denotes the integrated peak area ratio of $C_2H_4/C_2H_6$ detected by mass spectrometer. **c** Comparison of production rate of $C_2H_6$ before and after the addition of ethene in syngas feedstock, and (**d**) Effect of the ethene co-feeding upon the formation rate of the hydrocarbons products based on a carbon basis at 533 K and 0.5 MPa.

determined using in situ DRIFTS shows that a larger amount of remaining $CO_{ad}$ species with a lower intensity of $CH_4$ signal at approximately 3015 cm$^{-1}$ were observed for Na-Ru/SiO$_2$ (Fig. 3a and Supplementary Fig. 16). This result revealed that the Na promoter increases the strength of CO adsorption while hindering the reactivity of chemisorbed $H_2$, which can hydrogenate the surface carbon species obtained by $CO_{ad}$ dissociation to form $CH_4$. Similarly, the observed boosted peak intensity and higher formation temperature of $CH_4$ in CO-TPSR result confirm this point (Supplementary Fig. 17). An ethene pulse transient hydrogenation experiment was further performed. The catalysts were firstly reduced in $H_2$ for 2 h and reacted in syngas for 1 h, followed by switching to a $H_2$ flow at 533 K. The ethene was then pulsed into the systems and the effluent (ethane or ethene) was detected by mass spectrometer. As shown in Fig. 3b, an obvious larger ethane pulse peak was observed for Ru/SiO$_2$ with the peak area ratio of ethene to ethane (R) reached 5.5. However, the formation of ethane was almost totally inhibited for Na-Ru/SiO$_2$ case with R value surged to 22.5, in accordance with the observed phenomenon for propene pulse transient hydrogenation experiment (Supplementary Fig. 18). It was thus suggested that the Na promoter might increase the "inertness" of adsorbed H and suppress the secondary hydrogenation of olefins. The dynamic of chemisorbed $H_2$ can be further distinguished between Ru/SiO$_2$ and Na-Ru/SiO$_2$ via the experiment of co-feeding of ethene with syngas at 533 K and 0.5 MPa. As shown in Fig. 3c, a considerable part of added ethene was hydrogenated to ethane with formation rate increased by a factor of 592.8 and the ethene/ethane ratio was as low as 0.8 for Ru/SiO$_2$. However, the ethene readily desorbed with ethene/ethane ratio reaching 113.9 for Na-Ru/SiO$_2$ (Supplementary Fig. 19). Infrared studies of $C_2H_4$ adsorbed also confirmed the promotional effect of Na on the desorption of $C_2H_4$ at 533 K (Supplementary Fig. 20). Furthermore, the adsorption energy was calculated to be

−1.09 eV when ethylene was chemisorbed on top of Ru in the π mode, and the C = C bond length was calculated to be 1.45 Å (Supplementary Fig. 21). By comparison, the adsorption energy of ethylene was predicted to be −0.75 eV upon introducing Na$_2$O, and the C = C bond length was shortened to 1.43 Å, indicating that the interaction between ethylene and the Ru surface becomes weaker. It was suggested that Ru could acquire additional electrons from Na, and thus favored the desorption of ethylene as well as suppression of the possible secondary hydrogenation of olefins. In addition, according to catalytic performance with ethene co-feeding as shown in Fig. 3d, the absence of Na promoter in Ru/SiO$_2$ can boost the extent of ethene participation in the chain initiation and propagation to form $C_{3+}$ products with the increased formation rate of $C_3$-$C_5$ hydrocarbons by a factor of ~1.5, while an ignorable effect was found for Na-Ru/SiO$_2$ due to the weaker ethene adsorption ability on Na-modified Ru metal surface. Obviously, the Na doping would significantly decrease the sticking coefficients and reactivity of $H_2$ on Ru metal surface[23]. Previous works[22,23] has suggested that the alkali promoter would preferentially occupy the low-coordination corner and edge Ru sites required for the dissociation of chemisorbed $H_2$ using dynamic $^1H$ NMR spectroscopy. Therefore, the structure sensitive, highly mobile and weakly bound β-state of adsorbed $H_2$ disappeared for Na-promoted Ru NPs, which can exchange with the mobile α-state and gas phase $H_2$ in a fast way. Furthermore, the above characterization results also suggest that CO-rich Ru surface can be obtained for Na-Ru/SiO$_2$ catalyst due to the stronger electronic effect of Na promoter. These strongly adsorbed CO* molecules will occupy a large amount of exposed Ru metal sites and thus lower the surface coverage of $H_2$[25]. Such promotional effect of electron-rich metal center caused by doping alkaline promoter is also commonly observed for Fe- or Co-based catalysts for FTO reaction[10,11,15,17,32]. Based on these discussions, it is reasonable to

speculate that the Na promoter could decrease the fraction of Ru surface sites available for $H_2$ adsorption and reduce the mobility of chemisorbed $H_2$, thus decreasing the reactivity of $H_2$ and hydrogenation capacity of Ru-based FTS catalysts and rending them very efficient for unsaturated olefins production with limited $CH_4$ selectivity. In addition, the suppressed reactivity of chemisorbed $H_2$ may also lead to the decrease in catalytic activity.

Another intriguing result for the Na-Ru/$SiO_2$ catalyst is its strong tendency in hindering the formation of $CO_2$ with selectivity normally less than 3.0%, which has long been regarded as a great challenge for traditional STO catalytic systems[11]. In view of the ultralow intrinsic WGS reactivity of metallic Ru, we inferred that the Na-promoted Ru catalyst with metallic Ru as active phase possessed similar property. To verify this viewpoint, a WGS reaction probe experiment was performed and shown in Supplementary Fig. 22. After reaching a steady-state FTO performance, $H_2O$ stream was introduced into the working reactor. It was found that the TOF value for CO conversion and $CO_2$ selectivity remained almost unchanged with the value of ~ 0.102 s$^{-1}$ and 2.1%, respectively. Subsequently, $H_2$ flow was turned off, leaving CO and $H_2O$ as feedstock. Surprisingly, the TOF decreased evidently to 0.005 s$^{-1}$, suggesting that the $CO_2$ footprint produced via WGS reaction route can be almost disregarded. Furthermore, the Na doping also slightly increased the $CO_2$ selectivity for Ru/$SiO_2$ catalyst (Fig. 1f). Prior studies have revealed that the H-assisted $CO_{ad}$ dissociation route prevails on Ru cluster surface with near-saturation $CO^*$ coverage during FTS process[26], which features the preferential formation of $H_2O$ instead of $CO_2$ as the primary oxygen removing pathway (Supplementary Fig. 23), similar to those observed over metallic Co-based catalysts[15]. The suppressed reactivity of chemisorbed $H_2$ and increased CO adsorption might slightly benefit the generation of $CO_2$ after introducing Na to the Ru/$SiO_2$ catalyst.

### Industry-relevant testing

To expand maximumly the industrial application potential of the Na-Ru/$SiO_2$ catalyst, the utilization efficiency of noble Ru metal should be further improved. We found that the addition of polyvinylpyrrolidone (PVP) during the catalyst preparation procedure can greatly increase the Ru metal dispersion with higher exposed metallic surface area (Supplementary Table 7), which will significantly boost the production of olefins from syngas for a specific amount of Ru (Supplementary Table 9). The added PVP was completely removed during the calcination process (Supplementary Fig. 24), and the as-obtained catalysts were defined as Na-yRu(P)/$SiO_2$, where $y$ refers to the theoretical weight percent of Ru. Compared to Na-5%Ru/$SiO_2$, the reaction rate of CO over Na-5%Ru(P)/$SiO_2$ was significantly elevated to 0.702 mol$_{CO}$· g$_{Ru}^{-1}$·h$^{-1}$, which is about 1.5 times higher than the former (Supplementary Table 9). Particularly, the Na-2%Ru(P)/$SiO_2$ case with low concentration of Ru loading not only showed perfect catalytic stability (Fig. 1e), but also exhibited a better performance when being evaluated in a pilot-scale fixed-bed reactor (internal diameter: 19 mm; length: 1180 mm) under industrially relevant conditions. As shown in Supplementary Fig. 25, a TOF value of 0.312 s$^{-1}$ was acquired for the pellet catalyst with 72.5% of olefins selectivity together with 1.8% $CH_4$ selectivity and 2.5% $CO_2$ selectivity, which is similar to that evaluated in a microreactor and superior to most other Ru-based catalysts reported previously (Supplementary Table 10). Compared to previously industrial-favored Fe-based FTO catalyst, the Ru-based case simultaneously exhibits excellent olefins selectivity (~60% vs ~80%), olefins yield (~36% vs ~52%), and catalytic stability under very mild reaction conditions (i.e., 593-613 K vs 533 K, 2-3 MPa vs 1.0 MPa). Moreover, the Na-Ru/$SiO_2$ catalyst is more beneficial to the production of value-added long-chain olefins. Future researches are required to further decrease the Ru loading and catalyst cost from the viewpoint of economic feasibility.

## Discussion

In conclusion, we have developed a Na-promoted supported metallic Ru NPs catalyst that can effectively tune the dominated product distribution from traditional paraffins to value-added olefins with high selectivity and yield to justify the promising potential industrial application. The formation of C1 by-products including $CH_4$ and $CO_2$ is greatly suppressed within <5%. The as-obtained catalyst also significantly benefits the production of long-chain olefins with fraction in produced olefins surpassing 70%, and an excellent catalytic stability is also achieved. Given the technically available approaches for the recovery and recycling of the noble Ru metal, this work establishes brilliant prospect for direct production of olefins, especially for long-chain olefins, from syngas using Ru-based catalysts featuring ultrahigh carbon efficiency.

## Methods

### Catalysts preparation

All catalysts were prepared by the incipient wetness impregnation method. Typically, suitable amount of ruthenium nitrosyl nitrate solution (0.1414 g Ru per gram of solution, Heraeus Precious Metal Technology Co., Ltd.) was diluted with deionized water (13.5 mL) according to the required volume for incipient wetness impregnation of 5.559 g of aerosol silica ($SiO_2$, AEROSIL 380, Evonik Degussa China Co., Ltd.). Then different qualities of $NaNO_3$ (AR, Sinopharm Chemical Reagent Co., Ltd.) were added into the above ruthenium nitrosyl nitrate solution under vigorous stirring to obtain different Na/Ru molar ratios (0, 0.2, 0.5, 0.8, 1, and 2). Subsequently, the $SiO_2$ support was impregnated with the above precursor solution, followed by drying at 333 K for 4 h and calcination in air at 673 K for 4 h. The as-obtained sample was labeled as $x$Na-$y$Ru/$SiO_2$, whereas the $x$ denotes the molar ratio of Na/Ru, and $y$ denotes the theoretic weight of Ru loading. Generally, the sample with $x$ value of 0, and $y$ value of 5% was abbreviated as Ru/$SiO_2$, while the sample with $x$ value of 0.5, and $y$ value of 5% was abbreviated as Na-Ru/$SiO_2$. Specifically, according to above procedure, 2.122 g ruthenium nitrosyl nitrate solution with 0.126 g (1.484 mmol) of $NaNO_3$ can obtain the Na-Ru/$SiO_2$, while Ru-$SiO_2$ was obtained without the addition of $NaNO_3$. Both Ru/$SiO_2$ and Na-Ru/$SiO_2$ samples were mainly used for discussion in this work unless otherwise specified.

The same procedure was carried out for the synthesis of other alkali metal-promoted ruthenium catalysts with 5% of theoretic Ru loading and 0.5 of theoretic molar ratio of alkali promoter to Ru. Typically, 2.122 g ruthenium nitrosyl nitrate solution (0.1414 g Ru per gram of solution, Heraeus Precious Metal Technology Co., Ltd.) was diluted with 13.5 mL deionized water. 1.484 mmol of alkali metal nitrate ($LiNO_3$, $KNO_3$, $RbNO_3$, $CsNO_3$) were added into the above ruthenium nitrosyl nitrate solution under vigorous stirring, respectively. Subsequently, 5.559 g of $SiO_2$ support was impregnated with the above precursor solution, followed by drying at 333 K for 4 h and calcination in air at 673 K for 4 h. The as-obtained sample was labeled as M-Ru/$SiO_2$ (M=Li, K, Rb, Cs).

Moreover, for the PVP-assisted catalyst preparation, the sample was denoted as Na-$y$Ru(P)/$SiO_2$ catalyst, whereas the molar ratio of ANa/Ru was also fixed at 0.5, the $y$ denotes the theoretic weight of Ru loading, the P denotes the PVP ($M_w$ = 58000, Shanghai Aladdin Biochemical Technology Co., Ltd.), and the mass ratio of PVP/Ru was fixed at 10. In a typical synthesis procedure, a certain amount of polyvinylpyrrolidone (PVP) was dissolved in deionized water by vigorous stirring. Then, $NaNO_3$ and ruthenium nitrosyl nitrate solution were subsequently added to the dissolved PVP solution. The as-obtained mixed solution was then stirred for 12 h. After that, similar impregnation, drying, and calcination procedure were applied for the preparation of Na-$y$Ru(P)/$SiO_2$ as that of $x$Na-$y$Ru/$SiO_2$. Specifically, according to above procedure, 2.122 g ruthenium nitrosyl nitrate solution with

0.126 g (1.484 mmol) of $NaNO_3$ and 3.295 g PVP can obtain the sample of Na-5%Ru(P)/$SiO_2$.

## Catalytic evaluation

Catalysts were evaluated for syngas conversion in a continuous flow fixed-bed reactor with 10 mm inner and an inserted stainless-steel sleeve to monitor the reaction temperature. Typically, 1 g of catalyst sieved into 40 – 60 mesh was diluted with quartz sand (6 g, 40–60 mesh) and loaded into the constant temperature zone of the reactor. Prior to the catalytic reaction, the catalyst was reduced with pure $H_2$ (200 mL·min$^{-1}$) at 723 K for 4 h. After the reactor temperature was cooled down, a syngas with a $H_2$/CO ratio of 2/1 ($H_2$/CO/$N_2$ = 64.7/32.3/3) was introduced into the reactor at a flow rate of 50 mL·min$^{-1}$ (Weight hours space velocity (WHSV) = 3000 mL·g$_{cat.}^{-1}$·h$^{-1}$). The $N_2$ was used as internal standard to calculate the CO conversion and product selectivity in the tail gas. The reaction was carried out at 533 K, 3000 mL·g$_{cat.}^{-1}$·h$^{-1}$, 1.0 MPa, and $H_2$/CO ratio of 2 unless otherwise specified. After passing through a hot trap (393 K) and a cold trap (273 K), the gaseous effluent was analyzed online using an Agilent 7890B apparatus equipped with two detectors. A packed column (TDX-01) connected to a thermal conductivity detector (TCD) using He as carrier gas was used to analyze the $H_2$, $N_2$, CO, $CH_4$, and $CO_2$. A KCl-modified alumina capillary column (Agilent 19095P-K25) connected to a flame ionization detector (FID) using Ar as carrier gas was used to analyze the hydrocarbons with carbon number in the range of 1–7 ($C_1$-$C_7$). The aqueous products, liquid oil products, and solid wax products were collected from cold trap and hot trap, and then analyzed off-line with Shimadzu GC. The aqueous products were analyzed via two Porapak Q columns equipped with a TCD for the detection of $H_2O$ and MeOH and an FID for the detection of $C_1$-$C_5$ oxygenate. The liquid oil products were analyzed with an HP-1 column connected to an FID using $N_2$ as carrier gas. The wax product was dissolved in $CS_2$ and analyzed by an MXT-1 column with an FID using $N_2$ as carrier gas. The catalytic performance at the stable stage after 12 h of running based on gas analysis was typically used for discussion. The mass balance, carbon balance, and oxygen balance were calculated and maintained at 100 ± 5%. At least three repeated experiments carried out under the same reaction conditions demonstrated that the catalyst shows good reproducibility. Both CO conversion and product selectivity were calculated on a carbon-atom basis. The selectivity of oxygenates was less than 1%, and has been excluded from the reported product selectivity unless otherwise specified.

CO conversion ($X_{CO}$), product selectivity ($S_i$) and yield ($Y_{olefins}$) were calculated by the following equation:

$$X_{CO} = \frac{F_{co,in} - F_{co,out}}{F_{co,in}} \times 100\% \tag{1}$$

$$S_i = \frac{N_i \times n_i}{\sum (N_i \times n_i)} \times 100\% \tag{2}$$

$$Y_{olefins} = X_{CO} \times S_{olefins} \tag{3}$$

where $F_{co,in}$ and $F_{co,out}$ represent moles of CO at the inlet and the outlet, respectively, $S_i$ denotes the selectivity of product $i$ on a carbon basis, $N_i$ is the molar fraction of product $i$, and $n_i$ is the carbon number of product $i$, $Y_{olefins}$ denotes the yield of olefins product.

The reaction rate (R) and turnover frequency (TOF) of CO conversion were calculated using the following equations:

$$R_{CO} = \frac{V_{WHSV} \times X_{CO} \times N_{CO\,concentration}}{22400 \times \xi_{Ru}} \tag{4}$$

where $V_{WHSV}$ is the weight hourly space velocity (mL·g$_{cat.}^{-1}$·h$^{-1}$), $N_{CO\,concentration}$ denotes the molar concentration of CO in the feedstock, $\xi_{Ru}$

denotes the true loading of Ru measured by ICP-AES.

$$TOF_{CO} = \frac{R_{CO} \times M_{Ru}}{3600 \times D_{Ru}} \tag{5}$$

where $M_{Ru}$ is the atomic mass of Ru (101.07 g·mol$^{-1}$), and the $D_{Ru}$ denotes the dispersion of Ru metal measured by CO chemisorption results. The chain growth probability (α) was calculated according to Anderson-Schulz-Flory distribution:

$$\left(\frac{W_n}{n}\right) = (1 - \alpha)^2 \alpha^{(n-1)} \tag{6}$$

where $n$ is the carbon number of products, $W_n$ is the mass fraction of the hydrocarbons with a carbon number of $n$, and α is chain growth probability.

The equation of (6) can be rewritten as:

$$Ln\left(\frac{W_n}{n}\right) = (n - 1)Ln\alpha + 2Ln(1 - \alpha) \tag{7}$$

Plotting Ln ($W_n/n$) versus $n$ (carbon number), and the chain growth probability (α) can be obtained by calculating the slope (Lnα).

## Industry-relevant testing

For experiment in industry-relevant testing, 10 g of the pellet catalyst (Na-2%Ru(P)/$SiO_2$) with size in cylindrical shape of Φ 5.0 × 3.5 mm was firstly crushed and sieved into size of 12 – 20 mesh, and then diluted with 40 g of $SiO_2$, followed by loading into a pilot-scale fixed-bed reactor (internal diameter: 19 mm; length: 1180 mm), which was equipped for operation at industrial working conditions. Before reaction, the catalysts were reduced at 723 K for 4 h in a flow of pure $H_2$ with 1000 mL·min$^{-1}$. After that, the temperature was cooled down to 473 K, and the syngas ($H_2$/CO/$N_2$ = 64.7/32.3/3.0) with a flow rate of 500 mL·min$^{-1}$ was introduced into the reactor and then pressured to 1.0 MPa. The reaction conditions were as follows: 538 K, 1.0 MPa, 3000 mL·g$_{cat.}^{-1}$·h$^{-1}$, $H_2$/CO ratio of 2. The reaction effluent was analyzed with an Agilent 7890B apparatus equipped with two detectors after passing through a hot trap (393 K) and a cold trap (273 K). The liquid products and solid wax products were collected from the cold trap and hot trap, and then analyzed off-line with Shimadzu GC. The identic analysis methods were applied with that investigated in a fixed bed microreactor.

## Catalyst characterization

The Ru concentrations of various reduced samples were measured by inductively coupled plasma optical emission spectrometry (ICP-OES, Perkin Elmer).

Power X-ray diffraction (XRD) data were acquired using a Rigaku Ultima IV X-ray diffractometer (40 kV, 40 mA) equipped with Cu Kα radiation (λ = 1.54056 Å) with scanning angle from 10 to 90° at a scanning speed of 2°·min$^{-1}$. The identification of the structure phase was based on the JCPDS standard card. The XRD crystallite size was calculated by Scherer Formula.

Transmission electron microscopy (TEM) and high-solution transmission electron microscopy (HRTEM) images were obtained on an FEI Tecnai G2 F20 S-TWIN equipment with 200 kV accelerating voltage. Samples for (HR)TEM were prepared by dispersing the sample in ethanol followed by ultrasonication. The nanoparticle size distribution for each sample was determined using ~300 nanoparticles. High angle annular dark field scanning transmission electron microscopy (HAADF-STEM) images and the corresponding energy dispersion X-ray analysis (EDX) were conducted on a JEOL JEM-F200 microscope equipped with an Oxford EDX detector.

Hydrogen temperature-programmed reduction ($H_2$-TPR) was tested on a Micromeritics ChemiSorb2920 with a thermal conductivity detector (TCD). 50 mg of sample was pretreated with He (30 mL·min$^{-1}$) at 393 K for 1 h. After the temperature decreased to 323 K, a 5%$H_2$/Ar

(30 mL·min⁻¹) flow was introduced into the system, and the temperature was ramped from 323 K to 1073 K at a heating rate of 10 K·min⁻¹. The signal was recorded by a thermal conductivity detector.

X-ray photoelectron spectroscopy (XPS) was obtained by Thermo Fisher Scientific K-Alpha spectrometer equipped with the Al Kα (1486.6 eV) radiation source. Before XPS measurement, the sample was treated by ion etch to remove surface adsorbate and oxide layer. The results were calibrated by setting the C $1s$ peak of 284.8 eV.

CO chemisorption experiments were conducted on a Micromeritics ChemSorb2920 with a thermal conductivity detector (TCD). 150 mg catalyst was in situ pretreated with pure H₂ flow (40 mL·min⁻¹) at 723 K for 2 h and then purged with He for 30 min. After the reactor was cooled down to 323 K under He flow, a flow of 10% CO/He was dosed into the reactor until achieving saturated adsorption of CO. Dispersion of Ru was calculated by assuming the stoichiometry of CO/Ru to be 1/1.

In situ diffuse reflectance Fourier transform infrared spectroscopy (DRIFTs) of CO chemisorption experiments were carried out on a ThermoFisher Scientific FTIR spectrometer (Nicolet iS50) equipped with a mercury cadmium telluride (MCT) detector. 20 mg of sample was in situ reduced under a H₂ flow (40 mL·min⁻¹) for 2 h at 723 K. The gas flow was switched to Ar flow to purge the reaction cell for 0.5 h and then the reactor was cooled down to 323 K or 533 K. The background spectra were collected at 323 K or 533 K. Then, Ar flow was switched to CO flow until achieving saturated adsorption. Subsequently, Ar flow was again introduced into the system to remove the gaseous CO, and the DRIFT spectra of CO adsorption were thus collected. As for CO adsorbed (CO$_{ad}$) reactivity experiment at 533 K, after the collection of DRIFT spectra of CO adsorption at 533 K, a flow of diluted H₂ (10 mL·min⁻¹ H₂ and 30 mL·min⁻¹ Ar) was introduced into the cell system, and the DRIFT spectra were automatically recorded to acquire the reactivity of CO$_{ad}$ species. Additionally, the relative intensity of CO adsorbed was compared based on the changes of integrated peak areas at different reaction time at 533 K.

Thermal gravimetric (TG) analysis was performed on a NETZSCH TG/DTA instrument. About 20 mg sample was heated from room temperature to 1073 K in air (40 mL·min⁻¹) with 20 mL·min⁻¹ Ar (purge gas and shielding gas) at the heating rate of 10 K·min⁻¹.

Ethene adsorption experiment was carried out by diffuse reflectance Fourier transform infrared spectroscopy (DRIFTs). 20 mg of sample was reduced in situ under a H₂ flow (40 mL·min⁻¹) for 2 h at 723 K. The gas flow was switched to Ar flow to purge the reaction cell for 0.5 h and then the reactor was cooled down to 533 K. The background spectra were collected at 533 K under Ar atmosphere. Then, the Ar flow was switched to C₂H₄ flow until achieving saturated adsorption. Subsequently, the Ar flow was again introduced into the system to remove the gaseous C₂H₄, and the DRIFT spectra of C₂H₄ adsorption were thus collected.

X-ray absorption fine structure (XAFS) data was performed at the BL14W1 of Shanghai Synchrotron Radiation Facility (SSRF), China. The storage ring of the SSRF was operated at 3.5 GeV with a maximum current of 230 mA. All data were acquired at the Ru K-edge in transmission mode. X-ray absorption near-edge spectroscopy (XANES) and extended X-ray fine-structure (EXAFS) spectroscopy of samples were collected under ambient condition using a fixed-exit double-crystal Si (111) monochromator. The catalyst sample was pressed into pellets within LiF, and then placed inside a stainless steel in situ cell which was surrounded by a heater. Reduction of the Na-Ru/SiO₂ catalyst was carried out by heating the in-situ cell at 10 K/min in the following pure H₂ up to 573 K, during which the XAFS spectra were measured at 298 K, 423 K and 573 K, respectively. Then the reduced Na-Ru/SiO₂ catalyst was treated by syngas (H₂/CO = 2) in the in-situ cell at 533 K for 30 min and the XAFS spectra was collected. The data analysis was performed using IFEFFIT software package according to standard data analysis procedures[33]. The energy was

calibrated by collecting spectra of Ru foil standard sample. After appropriate background subtraction, the $k^2$-weighted EXAFS spectra of the Ru K-edge data ranges were assessed based on the quality of data generally between k = 3 – 12 Å⁻¹ and for R = 1 – 3 Å. All data fitting was performed by Artemis program in IFEFFIT. The value of the passive electron amplitude reduction factor, $S_0^2$, was determined to be 0.75 for Ru, by a fit of a reference Ru foil with a fixed coordination number of 12 to reflect the HCP structure of Ru.

C₂H₄ and C₃H₆ pulse transient hydrogenation experiments were performed on VDSorb-9Xi instruments equipped with a thermal conductivity detector (TCD) and a MKS Cirrus 2 mass spectrometer. 20 mg of catalyst was loaded into a U-tube reactor and in situ reduced with H₂ at 723 K for 2 h, and then pretreated under a flow of syngas (H₂/CO = 2) at 533 K for 1 h, followed by switching to a flow of 10%H₂/Ar (20 mL·min⁻¹). Subsequently, the C₂H₄ or C₃H₆ was pulsed into the system to complete the pulse transient hydrogenation. The effluent for C₂H₄ (m/z = 26) and C₂H₆ (m/z = 30), or C₃H₆ (m/z = 42) and C₃H₈ (m/z = 44) was monitored using a MKS Cirrus 2 mass spectrometer. The integrated peak area ratio of C₂H₄/C₂H₆ detected by mass spectrometer was calculated and denoted as R, which displays the hydrogenation capacity of different catalysts.

Water-gas-shift (WGS) reaction probe experiment was carried out in a continuous flow fixed-bed reactor with a 10 mm inner diameter. 1 g of Na-Ru/SiO₂ sample diluted by 6.0 g SiO₂ was loaded into the reactor and then pretreated by pure H₂ (50 mL·min⁻¹) at 723 K for 4 h. Then, the reactor was cooled down to 533 K, followed by switching a flow of syngas (50 mL·min⁻¹, H₂/CO/N₂ = 65/32/3, 3 vol.% of N₂ as internal standard) and then was pressurized to 1.0 MPa. After reaching a steady-state performance according to the analysis of gas phase, a water stream with 0.05 mL·min⁻¹ was pumped into the reactor by a high-pressure pump (P230, Dalian Elite Analytical Instruments Co., Ltd, China). The effluent was continuously monitored by an Agilent 7890B chromatograph. After reaching a steady-state performance, the H₂ flow was turned off, leaving only CO and water as feedstock. The catalytic performance was calculated based on the analysis of the chromatograph data.

Temperature-programmed surface reaction of CO (CO-TPSR) was performed on a Micromeritics ChemiSorb2920 with a thermal conductivity detector (TCD) and MKS Cirrus 2 mass spectrometer. 150 mg of sample was loaded into a U-type reactor and reduced in flow of H₂ at 723 K for 4 h, followed by flushing with He flow for 30 min. After cooling down to 323 K under He flow, a flow of CO was switched to achieve the saturated adsorption of CO. Subsequently, the gas CO was flushed out from the reactor by He flow. After that, the He stream was replaced by H₂ flow, and the reactor temperature was heated from 323 K to 773 K at a rate of 10 K·min⁻¹. The methane signal (m/z = 14) in the effluent gas was recorded by the mass spectrometer.

The co-feeding of ethene with syngas was employed in a continuous flow fixed-bed microreactor. The effluent gas was analyzed by a gas chromatograph equipped with a packed column (TDX-01) connected to a thermal conductivity detector (TCD) and a capillary column (19095P-K25) connected to a flame ionization detector (FID). 100 mg of Ru/SiO₂ (80–100 mesh) or 200 mg of Na-Ru/SiO₂ (80–100 mesh) was diluted with quartz sand (1 g, 80–100 mesh) and then loaded into the quartz tube. The sample was reduced at 723 K for 4 h in a flow of H₂ at atmospheric pressure. Then, the reactor was cooled down to the reaction temperature (533 K). Syngas (H₂/CO = 2) with flow rate of 30 mL·min⁻¹ and He with flow rate of 10 mL·min⁻¹ were introduced into the reaction system and the reactor pressure was also pressurized to 0.5 MPa. After reaching steady-state FTO performance, the He flow was switched to 10%C₂H₄/He (10 mL·min⁻¹). CO conversion and products selectivity were calculated by an Agilent 7890B apparatus. The reaction was carried out at low CO conversions (<2%) to ensure differential operation.

## Computational details

Periodic density functional theory (DFT) calculations were performed with the Vienna ab initio simulation package (VASP)[34,35]. The Perdew-Burke-Ernzerhof (PBE) exchange-correlation functional[36] was applied with a plane-wave basis set. A cutoff energy of 400 eV and the projector augmented wave (PAW) method was applied throughout the calculations[37]. A Gaussian smearing method with a width of 0.2 eV was used during iterative diagonalization of the Kohn-Sham Hamiltonian. The electronic energy of the structure was converged to $10^{-4}$ eV in the self-consistent field calculations, whereas the force on each relaxed atom was converged to 0.03 eV·Å$^{-1}$ in the ionic relaxation calculations. Both the dimer[38] and the climbing-image nudged elastic band (CI-NEB) method[39,40] were used to locate the transition states (TS), which were then verified to have one and only one imaginary frequency.

The lattice parameters of hexagonal Ru was calculated to be a = b = 10.82, c = 18.42, which were in good agreement with experimental[41] and other theoretical values[26]. The Ru (0001) surface with a p (4 × 4) supercell was employed in this work, which consists of 64 atoms in 4 layers. The bottom two layers were fixed to mimic the bulk, while the remaining layers, along with the adsorbates, were allowed to relax. A Γ-centered 3 × 3 × 1 Monkhorst–Pack k-point mesh and a 12Å vacuum layer perpendicular to the slab were used.

## Data availability

The data that support the findings of this study are available within the paper and its Supplementary Information, and all data are also available from the corresponding authors upon reasonable request. Source data are provided with this paper.

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

## Acknowledgements

This work was financially supported by the Natural Science Foundation of China (91945301 to L.Z., 22072177 to T.L., 22172188 to S.L.), National Key R&D Program of China (2021YFF0500702 to L.Z.), Natural Science Foundation of Shanghai (22JC1404200 to L.Z., 21ZR1471700 to T.L.), Program of Shanghai Academic/ Technology Research Leader (20XD1404000 to L.Z.), Key Research Program of Frontier Sciences, CAS (Grant No. QYZDB-SSW-SLH035 to L.Z.), the "Transformational Technologies for Clean Energy and Demonstration", Strategic Priority Research Program of the Chinese Academy of Sciences (Grant No. XDA21020600 to L.Z.), Youth Innovation Promotion Association of CAS. Specially, we acknowledge the XAFS station (BL14W1) of the Shanghai Synchrotron Radiation Facility for the XAS test.

## Author contributions

L.Z., Y.S. conceived and supervised the project, designed the study. H.Y, C.W. and T.L. performed most of the experiments. Y.A. and F.Y. conducted some catalytic evaluation experiments. Y.W., F.S., Z.J. performed the X-ray adsorption spectroscopy characterization and analysis, Y.W, Q.C, S.L, performed the DFT calculation, T.L., H.Y., C.W., L.Z. wrote and revised the paper. All authors discussed the results and commented on the manuscript.

## Competing interests

The authors declare no competing interests.
