## [Peer Review File · Nature Communications]

Title: Direct production of olefins from syngas with ultrahigh carbon efficiencyREVIEWER COMMENTS

Reviewer #1 (Remarks to the Author):

In this manuscript, the authors developed an active Na-Ru/SiO₂ catalyst for production of olefins from syngas. An enhanced olefin selectivity with limited selectivity of CH₄ and CO₂ was obtained by using the Ru-based catalyst, which usually favored the production of long-chain saturated hydrocarbons. The addition of sodium made a significant change in product selectivity, which was probably originated from the hydrogenation ability. The authors employed various characterizations and probe experiments to verify the hypothesis. This is an interesting work with good organization. After some revisions, this work can be published in Nature Communications.

Below are some suggestions:

1. In this work, the authors mainly compared the performance between Ru/SiO₂ and Na-Ru/SiO₂, but stability test was conducted on Na-2%Ru(P)/SiO₂. The authors would better give an explanation to the employment of Na-2%Ru(P)/SiO₂ the first time it appeared in Fig 1E. Besides, what about the stability of Ru/SiO₂?
2. The Ru/SiO₂ catalyst exhibited an enhanced activity than that of Na-Ru/SiO₂. The authors are required to add some explanation to the varied activity along with their structure. The activity results should also be added for better comparison in Fig S3.
3. H₂-TPR results suggested that the addition of Na retarded the reduction of RuO₂, but from XAS and XPS spectra, the Na-Ru/SiO₂ showed a slightly higher reduction degree of Ru than Ru/SiO₂. Did the H consumption vary with sodium addition? The authors would better give some explanations to the higher electron density in Na-Ru/SiO₂.
4. From the experimental section, I found that XPS spectra were ex-situ conducted. Why there was no Ru^{δ+} detected as shown in Fig 2E?
5. When compared with previously reported catalysts, the authors should pay attention to the reaction conditions. The authors would better provide the performance of Na-Ru/SiO₂ conducted at different conditions. And reaction rates of the catalysts should be included in Table S1.
6. A DFT simulation for olefin production will greatly improve the quality of this work.

Minor suggestions:

1. Generally, there are no citations included in the abstract.
2. The font sizes in a figure should be kept the same. Pay attention to the font sizes in Fig 2 and Fig 3.
3. Symbols used in this manuscript should be carefully checked, such as "mL", "C 1s", etc.
4. et al.

Reviewer #2 (Remarks to the Author):

Achieving the selective production of specific valuable hydrocarbons directly from syngas at high CO

conversion rates and high carbon efficiency is one of the biggest challenges in syngas chemistry. The excellent, high-quality work by Yu et al. reported in this manuscript represents a significant step forward in this direction. The authors employed a novel Na-promoted Ru/SiO₂ catalyst to realize the direct conversion of syngas to olefins (STO), especially to long chain C₅₊ olefins, with an outstanding olefins selectivity of up to 80.1% and a remarkably ultralow selectivity (< 5%) to unwanted C₁ by-products (CH₄, CO₂) at relatively high CO conversion (> 45%). The Na-Ru/SiO₂ catalyst features high stability with no signs of deactivation during at least 550 h on stream and exhibits excellent catalytic performance when tested in the form of cylindrical-shaped pellets under industrially relevant conditions and therefore with good prospects for practical application. The outstanding STO performance displayed by the Na-Ru/SiO₂ catalyst is ascribed to an increased electronic density of surface metallic Ru atoms due to electron transfer from the Na promoter (supported by XPS, CO-DRIFTS, and DFT) strengthening the CO adsorption capacity of the Ru sites, along with a reduced surface mobility of chemisorbed H species suppressing secondary hydrogenation of olefins. A further differential aspect of the Na-Ru/SiO₂ catalyst with respect to most current STO catalysts is its negligible activity for the competing WGS reaction resulting in an unusually ultralow CO₂ selectivity (< 3%) even at CO conversions as high as 68%. The discussions and conclusions of this study are well supported by experiments and advanced spectroscopic and microscopy characterizations of the catalysts at different stages.

Overall, this is an excellent work deserving publication in Nature Commun. I only have a few comments that the authors should address before its definitive acceptance for publication.

1. As correctly stated by the authors, further studies would be required to achieve a more effective use of Ru in the Na-Ru/SiO₂ catalyst in order to decrease its cost and thereby to enhance its economic viability. In this respect, I am wondering why the catalyst was reduced at the high temperature of 723 K in spite the H₂-TPR profiles indicate the complete reduction of RuO₂ to metallic Ru at temperatures below 500 K (Fig. S11). In principle, one might expect a higher Ru dispersion and therefore a more efficient metal utilization at lower reduction temperatures.
2. P10L227: "It was suggested that the Na promoter might increase the "internes" of adsorbed H and suppress the secondary hydrogenation of olefins". What does "internes" mean? Probably the authors meant "inertness". Please, clarify it.
3. P11L250: "Based on these discussions, it is reasonable to speculate (instead of "specular") that ...".
4. Methods. Please, check equation (3) for calculation of product yields. In my view it should be " $Y_i = XCO \times Si / 100$ " if both CO conversion (XCO) and selectivity (Si) are given in % according to equations (1) and (2).

Reviewer #3 (Remarks to the Author):

In this work, the authors presented Ru-based catalysts for the conversion of synthesis gas to olefins via

the Fischer-Tropsch to Olefins (FTO) technology. They highlighted the suppression of undesired C1 products (CO₂ and CH₄) to <5 % which leads to 80 % olefins selectivity at 46 % CO conversion. The robustness of the catalytic performance is thoroughly investigated, through the screening of various process parameters including temperature, pressure, H₂/CO ratios and space velocity. In addition to the steady-state catalytic performance, transient and co-feeding experiments were performed to probe the reactivity of the catalysts for ethene hydrogenation and water-gas-shift. Last but not least, the authors also utilized multiple analysis tools to identify catalyst properties which substantiated their findings. The authors deserve compliments for the high quality and detailed presentation of their results, and their meticulous design of experiments.

The direct conversion of synthesis gas to olefins facilitates a more sustainable production of chemicals from alternative feedstocks, leading to a circular carbon economy. Hence, a significant advancement would be valuable to the scientific community and society, warranting a spot in Nature Communication. In this case, the authors justify the importance of their work by reasoning that the olefins yields attained with their Na-promoted Ru/SiO₂ catalysts surpassed all state-of-art catalysts. The concept of suppressing C1 production is first proposed by Xie et al. (ref. 14) using Co-based catalysts and later by Xu et al. (ref. 10) using Fe-based catalysts so this is not new. Na promotion on Ru-based catalysts have been concluded to promote olefins production (ref. 21 and Williams and Lambert 2000, 10.1023/A:1019023418300) so this aspect is also supported by literature. However, there has been negligible progress on Ru-based catalysts, and this work brings awareness and encourages the exploration of Ru-based catalysts for olefins production with limited C1 production. Hence, this work is recommended for publication in Nature Communication if the following scientific points could be clarified/ improved.

1. Line 21 ‘... oligomerization of lower olefins lead to high value-added long-chain olefins via Ziegler-Natta polymerization process.’ The olefins produced is in the range of C₂-C₂₀, so the alternative commercial process should be the Shell Higher Olefins Process (e.g. Kiem 2013, 10.1002/anie.201305308) instead of Ziegler-Natta polymerization.
2. The significance of this work is arguably over-stated and over-simplified. Most state-of-art FTO catalysts focus on the selectivity of C₂-C₄ olefins and the olefins selectivity in the C₅+ products is not specified. However, this does not mean that the C₅+ fraction did not contain olefins, as the author inaccurately suggested in Figure 1a. For instance, the Co₁Mn₃-Na₂S catalysts showed <7 % C₁ products at 240 °C and 10 bar with olefin/paraffin ratio of ~ 5 (Figure 3a in ref. 14). This suggests that the olefins selectivity of the Co₁Mn₃-Na₂S catalysts (~70 %) is closer to the Na-Ru/SiO₂ catalysts than the authors depicted in Figure 1a and stated in line 68-69. The authors are recommended to go through the literature in detail to make an accurate comparison. Another point is that although the selectivity towards olefins was high, the carbon distribution remained broad due to ASF so the selectivity of each olefins was ≤10 %. Hence, it is perhaps too general to state that the C₅+ olefins are of value. Instead, the authors could specify the selectivity towards certain fraction of value, e.g. C₂-C₄ olefins for bulk chemicals, C₁₂-C₁₈ for detergents (Kiem 2013).
3. Stability tests shown in Figure 1e and S1 are for Na-2%Ru(P)/SiO₂ catalyst but all other catalyst test results are compared using Na-Ru/SiO₂ and Ru/SiO₂ catalysts. Representative stability tests for Na-Ru/SiO₂ and Ru/SiO₂ catalysts should be added.
4. In FTS and FTO processes, product selectivity is dependent on conversion so product selectivity should

be compared at similar conversion levels. For Co-based and Ru-based catalysts, CH₄ selectivity is shown to increase when CO conversion is >80 % (Yang et al. 2014, 10.1016/j.apcata.2013.10.061). Conversion results should be added to Figure 1 and S3 to demonstrate that the lower C1 selectivity attained by Na-Ru/SiO₂ was not due to a difference in conversion.

5. Line 178-179 'Specially, the homogenous distribution of Na over the catalyst surface may benefit the strong electronic interaction between Ru NPs and Na.' This is doubtful, because one would think that Na has to be in contact with the Ru NPs (i.e. Figure S10) to promote the Ru active sites and the Na species on the support acted as spectators. From the various characterization data, could the authors (semi)quantify the amount of Na on the support vs. on Ru NPs? This would clarify the role of Na when it is on the support vs. Ru NPs. The authors are further suggested to include HAADF-STEM images and EDX elemental mapping of the spent Ru/SiO₂ and 2Na-Ru/SiO₂ catalysts so as to check for the sensitivity of the Na signal and to prove the above sentence. The experimental loading of Na measured by ICP-OES should also be included.

6. Line 198-199 'Based on the linear characteristic of ASF distribution, we can infer that the surface carbide mechanism ...' This is not so accurate, because the linearity of the ASF distribution is a characteristic of the FTS and FTS technologies, regardless of surface carbide/ bulk carbide/ CO insertion mechanisms etc. The authors could perhaps refer to computational studies on Ru-based FTS catalysts by the group of Hensen (10.1039/c4cy00483c and 10.1002/anie.201406521) to strengthen their mechanism discussion.

7. Line 265-268 regarding the discussion on the WGS activity of Na-Ru/SiO₂. The authors failed to acknowledge that metallic Ru-based FT catalysts, similar to the metallic Co-based catalysts (in ref. 14), have negligible WGS activity at low/ moderate CO conversion levels. In Figure 1f, Ru/SiO₂ had no WGS activity and the WGS activity actually increased with increasing Na loading. The authors should clarify that the WGS activity increased with Na promotion and provide a possible explanation on why Na promotion increased WGS activity and what are the possible implications.

8. Line 471-472 regarding the scale-up operation. In the microreactor, plug-flow conditions appear to be fulfilled (reactor inner diameter >10 times catalyst particle sieve fraction, catalyst bed height >50 times catalyst particle sieve fraction). However, this does not appear to be the case for the pilot-scale reactor as the reactor inner diameter was only 4 to 6 times the dimensions of the extrudate. To demonstrate the success of the scale-up operation to claim 'industry-relevant testing', the pilot-scale reactor should also be operated under plug flow conditions and a direct comparison of the Na-2%Ru(P)/SiO₂ catalyst performance in the microreactor and the pilot-scale reactor would be appreciated.

9. Spelling mistakes: line 78 ('predicated' = predicted), line 250 ('specular' = speculated), line 421 ('stand' = standard)

10. Experimental methodology for x-ray spectroscopies is missing?

Reviewer report by Jingxiu Xie

Point-by-point responses to all the comments from referees

Reviewer #1

Comments:

In this manuscript, the authors developed an active Na-Ru/SiO₂ catalyst for production of olefins from syngas. An enhanced olefin selectivity with limited selectivity of CH₄ and CO₂ was obtained by using the Ru-based catalyst, which usually favored the production of long-chain saturated hydrocarbons. The addition of sodium made a significant change in product selectivity, which was probably originated from the hydrogenation ability. The authors employed various characterizations and probe experiments to verify the hypothesis. This is an interesting work with good organization. After some revisions, this work can be published in Nature Communications.

Below are some suggestions:

Author reply:

Thanks a lot for your valuable comments. The point-by-point responses to your comments are shown as follows:

Comments:

1. In this work, the authors mainly compared the performance between Ru/SiO₂ and Na-Ru/SiO₂, but stability test was conducted on Na-2%Ru(P)/SiO₂. The authors would better give an explanation to the employment of Na-2%Ru(P)/SiO₂ the first time it appeared in Fig 1E. Besides, what about the stability of Ru/SiO₂?

Author reply: Thank you for your professional comment. The catalytic stability of Ru/SiO₂ and Na-Ru/SiO₂ catalysts with 5 wt.% of Ru were compared and showed in Supplementary Fig. 2. Both Ru/SiO₂ and Na-Ru/SiO₂ catalysts showed high stability within 50 h of test. To better illustrate the outstanding FTO catalytic performance of Na promoted Ru-based catalyst, the preparation method was optimized, and a Na-2%Ru(P)/SiO₂ catalyst with a relative lower Ru loading amount was prepared for FTO evaluation. Polyvinylpyrrolidone (PVP) was used during catalyst preparation procedure to greatly increase the Ru metal dispersion with higher exposed metallic surface area. It was found that the Na-2%Ru(P)/SiO₂ catalyst showed high stability for 500 h, and olefins selectivity kept in the range of 75~80% while that of the undesired C1 by-products was always suppressed within 5%.

In the revised manuscript, the stability test of Ru/SiO₂ and Na-Ru/SiO₂ catalyst was added as *Supplementary Fig. 2 in the Supplementary Information*, and the corresponding explanation was added in the main text as following:

“Furthermore, stability test was carried out as shown in **Supplementary Fig. 2**. The catalytic performance for both Ru/SiO₂ and Na-Ru/SiO₂ catalysts remained stable within 50 h of test. Especially, the Na-2%Ru(P)/SiO₂ catalyst with much lower Ru loading amount (1.8 wt.% Ru, ICP) exhibited high stability for 500 h without any significant loss in activity and selectivity. Overall, the activity remained at around 0.700 mol_{CO}·g_{Ru}⁻¹·h⁻¹ with intrinsic TOF of 0.210 s⁻¹, and olefins selectivity in total products kept in the range of 75~80% while that of undesired C1 by-products was always suppressed within 5% (**Fig.1E** and **Supplementary Fig. 3**).”

Supplementary Fig. 2 | Catalytic stability of Ru/SiO₂ and Na-Ru/SiO₂. Reaction conditions: 533 K, 3000 mL·g_{cat}⁻¹·h⁻¹, 1 MPa, H₂/CO ratio of 2.

2. The Ru/SiO₂ catalyst exhibited an enhanced activity than that of Na-Ru/SiO₂. The authors are required to add some explanation to the varied activity along with their structure. The activity results should also be added for better comparison in Fig S3.

Author reply: Thanks for your valuable comment. The catalyst activity did change significantly after the introduction of Na promoter. Such phenomena are also commonly observed for metallic Co-based FTS catalyst, and it is always believed that the alkali metal exhibits negative effect on FTS activity. Several explanations have been put forward for the behavior of alkali promoter, i.e., electronic effect, site blocking effects,

and blocking of specific sites of importance for hydrogen adsorption and dissociation (*Catal. Today*, 2013,215, 60; *Chin. J. Chem.* 2017, 35, 918). In this work, the detailed structure difference between Ru/SiO₂ and Na-Ru/SiO₂ were characterized for comparison study. Structure characterization by XRD/HRTEM/XAFS/XPS in Fig. 2 and Supplementary Fig. 7 - 10 suggested that the Na doping did not change the Ru phase, and metallic Ru phase exclusively existed for both Ru/SiO₂ and Na-Ru/SiO₂. However, a significant electronic effect was observed after introducing Na promoter according to XAFS/DRIFTS/XPS, and an electron-rich Ru centers for Na-Ru/SiO₂ sample was observed, along with the observation of increased CO adsorption strength. The strongly adsorbed CO* molecules will occupy a large amount of exposed metallic Ru sites and thus lower the surface coverage of H₂. Specially, the experiment of H₂-TPR, in-situ CO-DRIFTS, CO-TPSR, C₂H₄-H₂-pulse and C₂H₄-cofeeding experiments in Fig. 3, Supplementary Fig. 14 - 18 confirmed that the Na promoter would significantly weaken the reactivity and mobility of chemisorbed hydrogen on Ru surface, thus hindering the C-O activation. These discussions were presented in Paragraph 4 of “*Results and discussion*”.

Moreover, according to the comment of reviewer, the activity results were added into the Fig. S6 in the revised supplementary information.

In the revised manuscript, some sentences were revised for better understanding in Paragraph 4 of “*Results and discussion*”:

“We speculate that the possible reason for the huge difference in catalytic performance of Ru/SiO₂ and Na-Ru/SiO₂ **including activity and selectivity** may lie in the discrepancy of dynamic of chemisorbed hydrogen”;

“Based on these discussions, it is reasonable to speculate that the Na promoter could decrease the fraction of Ru surface sites available for H₂ adsorption and reduce the mobility of chemisorbed H₂, thus decreasing the reactivity of H₂ and hydrogenation capacity of Ru-based FTS catalysts and rendering them very efficient for unsaturated olefins production with limited CH₄ selectivity. **In addition, the suppressed reactivity of chemisorbed H₂ may also lead to the decrease in catalytic activity.**”

Supplementary Fig. 6 | Catalytic performance of Ru/SiO₂ (A) and Na-Ru/SiO₂ (B) at various reaction temperatures.

3. H₂-TPR results suggested that the addition of Na retarded the reduction of RuO₂, but from XAS and XPS spectra, the Na-Ru/SiO₂ showed a slightly higher reduction degree of Ru than Ru/SiO₂. Did the H consumption vary with sodium addition? The authors would better give some explanations to the higher electron density in Na-Ru/SiO₂.

Author reply: Thanks for your valuable comment. According to the reviewer's comments, we calculated the apparent H₂ consumption through the H₂-TPR results. As shown in Table 1(for response), the addition of Na slightly increased the apparent H₂ consumption. The apparent hydrogen consumption was calculated according to the following equation (*Journal of Catalysis, 1988, 111(1): 59-66.*)

$$\text{Apparent hydrogen consumption} = \text{Reduction} + \text{Adsorption} + \text{Spillover} - \text{Desorption of chemisorbed hydrogen} - \text{Desorption of spillover hydrogen}$$

In the view of the complexity of the factors involved and the significant H₂ spillover-effect of Ru metal, it is difficult to calculate the actual H₂ consumption for reduction process alone. The increased apparent H₂ consumption for Na-Ru/SiO₂ was mainly caused by the improved metallic Ru dispersion (Supplementary Table 7) and decreased metallic Ru particle size (Fig. 2B, Supplementary Fig. 8) after introducing Na promoter. Smaller Ru particle size with abundant edges and corners site could adsorb much more amount of H₂.

Table 1 (for response) The H₂ consumption calculation results.

Sample	Theoretical H ₂ consumption H _T ($\mu\text{mol}\cdot\text{g}^{-1}$)	Measured H ₂ consumption H _M ($\mu\text{mol}\cdot\text{g}^{-1}$)
Ru/SiO ₂	905.0	681.6
Na-Ru/SiO ₂	827.7	735.4

In this work, all catalysts were reduced at 723 K for 4 hours under a flow of H₂ before reaction, and both Ru/SiO₂ and Na-Ru/SiO₂ could be reduced completely under above reduction conditions according to H₂-TPR profiles. From XAS and XPS spectra, the exclusive metallic Ru species in both Ru/SiO₂ and Na-Ru/SiO₂ were identified. The slight shift in XANES was due to the electron-donor effect of sodium. The increased electronic density of Ru center for Na-Ru/SiO₂ can clearly be confirmed by XPS and CO-DRIFTS spectra, as shown in Fig. 2E, respectively. Furthermore, the Bader charge analysis based on DFT calculations demonstrated the electron-rich of surface Ru metal species after loading Na ion on the Ru (0001) surface (Supplementary Fig. 13), which strengthened CO adsorption and resulted in a CO-rich and H₂-poorn local chemical environment that benefits FTO performance.

4. From the experimental section, I found that XPS spectra were ex-situ conducted. Why there was no Ru^{δ+} detected as shown in Fig 2E?

Author reply: Thanks for your professional comment. As indicated by the reviewer, the XPS spectra were collected under ex-situ condition. Actually, a passivation treatment with 1 vol% O₂/Ar was specially performed for these catalysts. In order to remove adsorbate and oxide layer, we employed the argon ion etching method during the ex-situ XPS experiment. We are sorry that the relative experimental detail was not presented in the original manuscript. We have revised it and more details were added in the part of experimental *methods*.

In addition, we compared the XPS spectra of sample without argon ion etching treatment process. The corresponding fitting spectra results are shown in Fig.1(for response). The presence of an oxidation state of Ru species was observed on the reduced

Ru/SiO₂ and Na-Ru/SiO₂ catalysts surface due to passivation treatment, and the peaks at 280.9 eV and 280.7 eV were assigned to Ru^{δ+} species (*J. Mater. Chem.* 2012, 22, 14944-14950; *J. Am. Chem. Soc.* 2018, 140, 4172-4181).

Fig. 1 (for response) The fitting XPS spectra results of reduced Ru/SiO₂ and Na-Ru/SiO₂ catalysts without argon ion etching treatment process.

In the revised manuscript, the detailed XPS experiment method was added:

“X-ray photoelectron spectroscopy (XPS) was obtained by Thermo Fisher Scientific K-Alpha spectrometer equipped with the Al K α (1486.6 eV) radiation source. Before XPS measurement, the sample was treated by argon ion etching to remove surface adsorbate and oxide layer. The results were calibrated by setting the C 1s peak of 284.8 eV.”

5. When compared with previously reported catalysts, the authors should pay attention to the reaction conditions. The authors would better provide the performance of Na-Ru/SiO₂ conducted at different conditions. And reaction rates of the catalysts should be included in Table S1.

Author reply: Thank you for your valuable comment. In Supplementary Table 1, the detailed reaction conditions for various catalysts during syngas conversation to olefins

(STO) reaction have been listed. As shown in Supplementary Table 1, the as-obtained Na-Ru/SiO₂ catalyst exhibited the highest selectivity and yield for olefins (especial for C₅₊) together with the lowest fraction of undesired C1 by-products including CH₄ and CO₂.

The catalytic performance of Na-Ru/SiO₂ conducted at different reaction conditions including ratios of H₂/CO (Fig. 1C), space velocities (Fig. 1D), reaction temperatures (Supplementary Fig. 6), and reaction pressures (Supplementary Table 4), has been presented in original manuscript. For better comparison, the detailed tabular data were compiled into *Supplementary Table 2, 3, 5*.

Supplementary Table 2. Effect of H₂/CO ratio on catalytic performance of Na-Ru/SiO₂.^a

H ₂ /CO ratio	CO Conv. (%)	Selectivity (C %)				Yield (%)
		Olefins	C ₂₊ paraffins ^b	CO ₂	CH ₄	
0.5	6.8	77.0	16.6	4.4	2.0	5.2
1	15.2	76.4	17.0	4.2	1.6	11.6
2	45.8	80.1	15.0	2.7	2.2	36.7
4	61.0	72.6	19.7	2.6	5.1	44.3
5	72.4	69.7	22.5	2.1	5.7	50.5

^a Reaction condition: 1 MPa, 533 K, 3000 mL · g_{cat.}⁻¹ · h⁻¹.

^b Paraffins with two or more carbon atoms.

Supplementary Table 3. Effect of space velocity on catalytic performance of Na-Ru/SiO₂.^a

WHSV (mL · g _{cat.} ⁻¹ · h ⁻¹)	CO Conv. (%)	Selectivity (C %)				Yield (%)
		Olefins	C ₂₊ paraffins ^b	CO ₂	CH ₄	
1500	67.9	76.6	16.7	2.7	4.0	51.9
3000	45.8	80.1	15.0	2.7	2.2	36.7
6000	13.1	78.3	14.3	3.0	4.4	10.2
9000	7.5	78.6	13.4	3.3	4.7	5.9

^a Reaction condition: 1 MPa, 533 K, H₂/CO=2.

^b Paraffins with two or more carbon atoms.

Supplementary Table 5. Effect of reaction temperature on catalytic performance of Na-Ru/SiO₂.^a

Temperature (K)	CO Conv. (%)	Selectivity (C %)				Yield (%)
		Olefins	C ₂₊ paraffins ^b	CO ₂	CH ₄	
493	13.8	69.3	27.6	0.4	2.7	9.6
513	26.2	69.9	28.9	0.8	3.0	18.3
533	45.8	80.1	15.0	2.7	2.2	36.7
553	60.9	73.9	18.2	5.0	2.9	45.0

^a Reaction condition: 1 MPa, H₂/CO=2, 3000 mL • g_{cat.}⁻¹ • h⁻¹.

^b Paraffins with two or more carbon atoms.

Moreover, according to the suggestion of Reviewer, the reaction rates of various catalysts were calculated and added in **Supplementary Table 1**.

Supplementary Table 1 | Comparison of catalytic performance with previous works.

Entry	Catalyst category	Catalyst	T (K)	P (MPa)	H ₂ /CO ratio	WHSV (mL·g _{cat.} ⁻¹ ·h ⁻¹)	CO Conv. (%)	Reaction Rate (mol·g _{cat.} ⁻¹ ·h ⁻¹)	Product Selectivity (%)					Olefins yield (%)	Ref.
									CO ₂	CH ₄	C1(CO ₂ +CH ₄)	Olefins	Others		
1	Oxide-Zeolite	ZnCrO ₃ /MSAPO	673	2.5	2.5	5143	17.0	0.0112	41.0	1.2	42.2	47.2 ^e	10.6	8.0 ^e	(3)
2		ZnZrO ₃ /SAPO	673	1	2	3600	9.5	0.0051	45.0	6.0	51.0	34.7 ^e	14.3	3.3 ^e	(4)
3		ZnCrO ₃ /MOR	673	2.5	2.5	1857	12.0	0.0028	45.0	2.8	47.8	44.0 ^e	8.2	5.3 ^e	(5)
4		ZnAl ₂ O ₄ /SAPO-34	663	4	1	12000	6.9	0.0185	33.1	3.7	36.8	51.5 ^e	11.7	3.6 ^e	(6)
5		MnO _x /SAPO	673	2.5	2.5	4800	8.5	0.0052	41.0	2.0	43.0	46.7 ^e	10.3	4.0 ^e	(7)
6	Fe-based	Fe-Zn-0.81Na	613	2	2.7 ^e	60000	77.2	0.5589	23.0	9.7	32.7	60.2	7.1	46.5	(8)
7		FeMn@Si-c	593	3	2	4000	56.1	0.0334	13.0	10.0	23.0	65.3	11.7	36.6	(9)
8		Fe/α-Al ₂ O ₃	613	2	1	1500 ^b	80.0	-	40.0	6.6	46.6	31.8 ^e	21.6	25.4 ^e	(10)
9		Fe-K/CNTs	573	0.1	1	4200	16.5	0.0155	23.6	17.3	40.9	41.7 ^e	17.4	6.9 ^e	(11)
10		Fe/hNCNC	623	0.1	1	12000	3.5	0.0094	39.4	25.0	64.4	32.8 ^e	2.8	1.1 ^e	(12)
11		Mn/Fe ₃ O ₄	593	1	1	4480	41.5	0.0415	37.8	9.7	47.5	37.4 ^e	15.1	15.5 ^e	(13)
12		Fe10In/Al ₂ O ₃	673	0.5	2	7800	11.0	0.0128	16.0	-22.0	-38.0	45.0 ^e	-17.0	5.0 ^e	(14)
13		Fe ₃ O ₄ @MnO ₂	553	2	1	3000	67.9	0.0455	47.1	3.6	50.7	41.9	7.4	28.5	(15)
14		CoMn	523	0.1	2	2000	31.8	0.0095	47.3	2.6	49.9	32.0 ^e	18.1	10.2 ^e	(16)
15		Co-based	Co ₃ Mn ₃ Na ₂ S	513	0.1	2	-	0.8	-	<3.0	17.0	<20.0	54.0 ^e	26.0	0.4 ^e
16	0.5Na/CoMnAl@6.6Si		533	1	0.5	4000	13.5	0.0161	16.7	4.3	21.0	61.1	17.9	8.2	(18)
17	1.0Pr-CoRu/AOmM		473	2	2	-	20±3	-	0.9	8.4	9.3	19.9 ^d	-	-	(19)
18	Ru-based	Na-5% Ru/SiO ₂	533	1	2	3000	45.8	0.0204	2.7	2.2	4.9	80.1	15.0	36.7	
19		Na-5% Ru/SiO ₂	533	1	2	1500	67.9	0.0152	2.7	4.0	6.7	76.6	16.7	51.9	This work
20		Na-5% Ru(P)/SiO ₂	533	1	2	3000	65.3	0.0292	2.7	1.9	4.6	73.7	21.7	48.1	

^a 8 C% of CO₂ is included in the syngas feedstock (CO:H₂:CO₂:Ar=24:64:8:4).

^b GHSV of 1500 h⁻¹ is used.

^c The values denote the selectivity and yield of lower olefins (C₂₋₄⁻).

^d C₅-C₁₁ olefins.

6. A DFT simulation for olefin production will greatly improve the quality of this

work.

Author reply: Thank you for your valuable comment. We agree with the reviewer that a DFT simulation for olefins production will greatly improve the quality of this work. It is well known that Fischer-Tropsch synthesis is a complex reaction and it remains a great challenge to elucidate each reaction step. According to the widely accepted surface carbide mechanism, the key steps for olefins production are the hydrogenation or β -hydride abstraction of surface intermediate to terminate the carbon chain growth. The secondary hydrogenation of olefins intermediates is another vital step for olefins production. Based on these analyses, we have calculated and compared the energies for ethylene adsorption on Ru (0001) and Na₂O/Ru (0001) surfaces. As shown in *Supplementary Fig. 20*, the ethylene adsorption energies on Ru (0001) and Na₂O/Ru (0001) surfaces were calculated to be -1.09 and -0.75 eV, respectively. Thus, the introduction of Na₂O significantly weakened the adsorption strength of Ru (0001) toward C₂H₄, consistent with our observations from C₂H₄-DRIFTS and C₂H₄-pulse, C₂H₄-cofeeding experiments. Therefore, the secondary hydrogenation of olefins intermediates can be greatly suppressed with the Na addition. More works about the detailed DFT calculations on the role of the alkali promoter on the Ru metal surface for the FTS reaction will be carried out in our following researches.

In the revised manuscript and supplementary information, the following sentences and figure (*Supplementary Fig. 20*) were added.

“Furthermore, the adsorption energy was calculated to be -1.09 eV when ethylene was chemisorbed on top of Ru in the π mode, and the C=C bond length was calculated to be 1.45 Å (*Supplementary Fig. 20*). By comparison, the adsorption energy of ethylene was predicted to be -0.75 eV upon introducing Na₂O, and the C=C bond length was shortened to 1.43 Å, indicating that the interaction between ethylene and the Ru surface becomes weaker. It was suggested that Ru could acquire additional electrons from Na, and thus favored the desorption of ethylene as well as suppression of the possible secondary hydrogenation of olefins.”

Supplementary Fig. 20 | Optimized geometries of ethylene on Ru (0001) and Na₂O/Ru (0001).

7. Minor suggestions:

- (1). Generally, there are no citations included in the abstract.
- (2). The font sizes in a figure should be kept the same. Pay attention to the font sizes in Fig 2 and Fig3.
- (3). Symbols used in this manuscript should be carefully checked, such as “mL”, “C 1s”, etc.

Author reply: Thanks for your valuable comment.

- (1) The abstract has been rewritten with no citations as follow:

“Syngas conversion serves as a competitive strategy to produce olefins chemicals from nonpetroleum resources. However, the goal to achieve desirable olefins selectivity with limited undesired C1 by-products remains a grand challenge. Herein, we present a non-classical Fischer-Tropsch to olefins process featuring high carbon efficiency that realizes 80.1% olefins selectivity with ultralow total selectivity of CH₄ and CO₂ (< 5%) at CO conversion of 45.8%. This is enabled by sodium-promoted metallic ruthenium (Ru) nanoparticles with negligible water-gas-shift reactivity. Change in the local electronic structure and the decreased reactivity of chemisorbed H species on Ru surfaces tailored the reaction pathway to favor olefins production. No obvious deactivation was observed within 550 hours and the pellet catalyst also exhibited

excellent catalytic performance in a pilot-scale reactor, suggesting promising practical applications.”

(2) We have corrected the font size in Fig. 2 and Fig. 3, and we also checked the other Figures and make sure the same font size was applied.

(3) We have checked the symbols used in manuscript carefully and addressed this issue in the revised manuscript and supplementary information.

Reviewer #2

Comments:

Achieving the selective production of specific valuable hydrocarbons directly from syngas at high CO conversion rates and high carbon efficiency is one of the biggest challenges in syngas chemistry. The excellent, high-quality work by Yu et al. reported in this manuscript represents a significant step forward in this direction. The authors employed a novel Na-promoted Ru/SiO₂ catalyst to realize the direct conversion of syngas to olefins (STO), especially to long chain C₅₊ olefins, with an outstanding olefins selectivity of up to 80.1% and a remarkably ultralow selectivity (< 5%) to unwanted C1 by-products (CH₄, CO₂) at relatively high CO conversion (> 45%). The Na-Ru/SiO₂ catalyst features high stability with no signs of deactivation during at least 550 h on stream and exhibits excellent catalytic performance when tested in the form of cylindrical-shaped pellets under industrially relevant conditions and therefore with good prospects for practical application. The outstanding STO performance displayed by the Na-Ru/SiO₂ catalyst is ascribed to an increased electronic density of surface metallic Ru atoms due to electron transfer from the Na promoter (supported by XPS, CO-DRIFTS, and DFT) strengthening the CO adsorption capacity of the Ru sites, along with a reduced surface mobility of chemisorbed H species suppressing secondary hydrogenation of olefins. A further differential aspect of the Na-Ru/SiO₂ catalyst with respect to most current STO catalysts is its negligible activity for the competing WGS reaction resulting in an unusually ultralow CO₂ selectivity (< 3%) even at CO conversions as high as 68%. The discussions and conclusions of this study are well supported by experiments and advanced spectroscopic and microscopy characterizations of the catalysts at different stages. Overall, this is an excellent work deserving publication in Nature Commun. I only have a few comments that the authors should address before its definitive acceptance for publication.

Author reply:

Thank you very much for your valuable comments. The point-by-point responses to your comments are shown as follows:

Comments:

1. As correctly stated by the authors, further studies would be required to achieve a more effective use of Ru in the Na-Ru/SiO₂ catalyst in order to decrease its cost and thereby to enhance its economic viability. In this respect, I am wondering why the catalyst was reduced at the high temperature of 723 K in spite the H₂-TPR profiles indicate the complete reduction of RuO₂ to metallic Ru at temperatures below 500 K (Fig. S11). In principle, one might expect a higher Ru dispersion and therefore a more efficient metal utilization at lower reduction temperatures.

Author reply: Thanks for your valuable suggestions. The utilization efficiency of noble Ru metal is a key factor that influences its future industrial application, and more studies are required for Na-Ru/SiO₂ catalytic systems. We agree with the reviewer that the reduction temperature plays a significant role in affecting the Ru dispersion and metal utilization efficiency. According to the hint of reviewer, the dispersion of metallic Ru for Na-Ru/SiO₂ catalyst under different reduction temperatures were calculated. As shown in Table 2(for response), a higher Ru dispersion was obtained at lower reduction temperature as expected.

Table 2(for response) Effect of reduction temperature on Ru dispersion degree.

Reduction temperature (K)	CO uptake ($\mu\text{mol}\cdot\text{g}^{-1}$)	Metallic Surface Area ($\text{m}^2\cdot\text{g}_{\text{Ru}}^{-1}$)	D _{CO} (%)
473	59.9	64.9	14.5
573	52.7	57.1	12.8
723	45.5	49.3	11.0

In this work, a reduction temperature of 723 K for all of samples was applied after referencing the most published literature (*Nat Commun.* 11, 3185 (2020); *Applied Catalysis B: Environmental* 278 (2020) 119261), and we mainly focused on explaining the essential reason for the huge difference in catalytic performance between Ru/SiO₂ and Na-Ru/SiO₂. In the follow-up research, we will continue to comprehensively study the effect of reduction temperature on structure-performance relationship of Na-Ru/SiO₂ cases. Strategies to improve the metal utilization efficiency is also a continuous

on-going research work.

2. P10L227: “It was suggested that the Na promoter might increase the “internes” of adsorbed H and suppress the secondary hydrogenation of olefins”. What does “internes” mean? Probably the authors meant “inertness”. Please, clarify it.

Author reply: Thanks for your valuable suggestions. We are sorry for the incorrect spelling, and we have corrected it as “inertness” in the revised manuscript.

3. P11L250: “Based on these discussions, it is reasonable to speculate (instead of “specular”) that ...”.

Author reply: Thanks for your valuable suggestions. We are sorry for the incorrect spelling, and we have corrected it as “speculate” in the revised manuscript.

4. Methods. Please, check equation (3) for calculation of product yields. In my view it should be “ $Y_i = X_{CO} \times S_i / 100$ ” if both CO conversion (X_{CO}) and selectivity (S_i) are given in % according to equations (1) and (2).

Author reply: Thanks for your valuable suggestions. We have revised the equation in the revised manuscript.

Reviewer #3

Comments:

In this work, the authors presented Ru-based catalysts for the conversion of synthesis gas to olefins via the Fischer-Tropsch to Olefins (FTO) technology. They highlighted the suppression of undesired C1 products (CO₂ and CH₄) to <5 % which leads to 80 % olefins selectivity at 46 % CO conversion. The robustness of the catalytic performance is thoroughly investigated, through the screening of various process parameters including temperature, pressure, H₂/CO ratios and space velocity. In addition to the steady-state catalytic performance, transient and co-feeding experiments were performed to probe the reactivity of the catalysts for ethylene hydrogenation and water-gas-shift. Last but not least, the authors also utilized multiple analysis tools to identify catalyst properties which substantiated their findings. The authors deserve compliments for the high quality and detailed presentation of their results, and their meticulous design of experiments.

The direct conversion of synthesis gas to olefins facilitates a more sustainable production of chemicals from alternative feedstocks, leading to a circular carbon economy. Hence, a significant advancement would be valuable to the scientific community and society, warranting a spot in Nature Communication. In this case, the authors justify the importance of their work by reasoning that the olefins yield attained with their Na-promoted Ru/SiO₂ catalysts surpassed all state-of-art catalysts. The concept of suppressing C1 production is first proposed by Xie et al. (ref. 14) using Co-based catalysts and later by Xu et al. (ref. 10) using Fe-based catalysts so this is not new. Na promotion on Ru-based catalysts have been concluded to promote olefins production (ref. 21 and Williams and Lambert 2000, 10.1023/A:1019023418300) so this aspect is also supported by literature. However, there has been negligible progress on Ru-based catalysts, and this work brings awareness and encourages the exploration of Ru-based catalysts for olefins production with limited C1 production. Hence, this work is recommended for publication in Nature Communication if the following scientific points could be clarified/improved.

Author reply:

Thank you very much for your valuable comments. The point-by-point responses to your comments are shown as follows:

1. Line 21 ‘... oligomerization of lower olefins lead to high value-added long-chain olefins via Ziegler-Natta polymerization process.’ The olefins produced is in the range of C₂-C₂₀, so the alternative commercial process should be the Shell Higher Olefins Process (e.g. Kiem 2013, 10.1002/anie.201305308) instead of Ziegler-Natta polymerization.

Author reply: We gratefully appreciate for your professional comment. According to your suggestion, the sentence of “via Ziegler-Natta polymerization process” has been deleted in the revised manuscript and references mentioned by the reviewers were cited.

2. The significance of this work is arguably over-stated and over-simplified. Most state-of-art FTO catalysts focus on the selectivity of C₂-C₄ olefins and the olefins selectivity in the C₅₊ products is not specified. However, this does not mean that the C₅₊ fraction did not contain olefins, as the author inaccurately suggested in Figure 1a. For instance, the Co₁Mn₃-Na₂S catalysts showed <7 % C₁ products at 240 °C and 10 bar with olefin/paraffin ratio of ~ 5 (Figure 3a in ref. 14). This suggests that the olefins selectivity of the Co₁Mn₃-Na₂S catalysts (~70 %) is closer to the Na-Ru/SiO₂ catalysts than the authors depicted in Figure 1a and stated in line 68-69. The authors are recommended to go through the literature in detail to make an accurate comparison. Another point is that although the selectivity towards olefins was high, the carbon distribution remained broad due to ASF so the selectivity of each olefins was ≤10 %. Hence, it is perhaps too general to state that the C₅₊ olefins are of value. Instead, the authors could specify the selectivity towards certain fraction of value, e.g. C₂-C₄ olefins for bulk chemicals, C₁₂-C₁₈ for detergents (Kiem 2013).

Author reply: We gratefully appreciate for your professional comment. We sincerely apologize for neglecting to label the selectivity to lower olefins (C₂₋₄⁺) in Fig. 1A. This

may cause some misunderstanding to the readers and reviewers. In Fig. 1A in the revised manuscript, we have made notation to these cases reporting lower olefins (C₂₋₄) selectivity.

Fig. 1A. Catalytic performance for direct syngas conversion to olefins. (A) Comparison of catalytic performance among Na-Ru/SiO₂ and other previously reported catalysts^{7-11,13,15}. (a: C₂₋₄ selectivity)

In addition, we agree with the reviewer that the importance of focusing on selectivity of certain carbon number of olefins. According to the data shown in Fig.1B, the olefins distribution with certain carbon number range is showed in *Supplementary Fig. 1*. We also added some description in the revised manuscript as follows:

“Specially, the fraction of value-added C₅-C₁₁ α -olefins was as high as 57.8%, which can be used for production of high-quality lubricant, plasticizer and surfactant, while the fraction of detergent-range C₁₂-C₁₈ α -olefins reached 16.4% (**Supplementary Fig. 1**).”

Supplementary Fig. 1 | Detailed olefins distribution of Na-Ru/SiO₂.

“Supplementary Notes:

The Na-Ru/SiO₂ catalyst exhibits a narrower carbon distribution compared with the classical FT catalysts. The fraction of lower olefins (C₂₋₄) accounts for 25.5%, which is commonly used for bulk chemicals. While the fraction of C₅-C₁₁ olefins reaches 57.8%, and can be widely used as raw materials and/or intermediates for production of chemicals such as lubricant, plasticizer and surfactant. In addition, the C₁₂-C₁₈ slate olefins with fraction of 16.4% favors the production of detergent.”

3. Stability tests shown in Figure 1e and S1 are for Na-2%Ru(P)/SiO₂ catalyst but all other catalyst test results are compared using Na-Ru/SiO₂ and Ru/SiO₂ catalysts. Representative stability tests for Na-Ru/SiO₂ and Ru/SiO₂ catalysts should be added.

Author reply: Thank you for your professional comment. The catalytic stability of Ru/SiO₂ and Na-Ru/SiO₂ catalysts with 5 wt.% of Ru were also compared as showed in *Supplementary Fig. 2*. Both Ru/SiO₂ and Na-Ru/SiO₂ catalysts showed promising stability within 50 h of test. To better illustrate the outstanding FTO catalytic performance of Na promoted Ru-based catalyst, the preparation method was optimized, and a Na-2%Ru(P)/SiO₂ catalyst with a relative lower Ru loading amount was prepared

for FTO evaluation. Polyvinylpyrrolidone (PVP) was used during catalyst preparation procedure to greatly increase the Ru metal dispersion with higher exposed metallic surface area. It was found that the Na-2%Ru(P)/SiO₂ catalyst showed high stability for 500 h, and olefins selectivity kept in the range of 75~80% while that of undesired C1 by-products was always suppressed within 5%.

In the revised manuscript, the stability test of Ru/SiO₂ and Na-Ru/SiO₂ catalyst was added as *Supplementary Fig. 2* in the Supplementary Information, and the corresponding explanation was added in the main text as following:

“Furthermore, stability test was carried out as shown in *Supplementary Fig. 2*. The catalytic performance for both Ru/SiO₂ and Na-Ru/SiO₂ catalysts remained stable within 50 h of test. Especially, the Na-2%Ru(P)/SiO₂ catalyst with much lower Ru loading amount (1.8 wt.% Ru, ICP) exhibited high stability for 500 h without any significant loss in activity and selectivity. Overall, the activity remained at around 0.700 mol_{CO}·g_{Ru}⁻¹·h⁻¹ with intrinsic TOF of 0.210 s⁻¹, and olefins selectivity in total products kept in the range of 75~80% while that of undesired C1 by-products was always suppressed within 5% (**Fig.1E** and **Supplementary Fig. 3**).”

Supplementary Fig. 2 | Catalytic stability of Ru/SiO₂ and Na-Ru/SiO₂. Reaction conditions: 533 K, 3000 mL·g_{cat}⁻¹·h⁻¹, 1 MPa, H₂/CO ratio of 2.

4. In FTS and FTO processes, product selectivity is dependent on conversion so product selectivity should be compared at similar conversion levels. For Co-based and Ru-based catalysts, CH₄ selectivity is shown to increase when CO conversion is >80 % (Yang et al. 2014, 10.1016/j.apcata.2013.10.061). Conversion results should be added to Figure 1 and S3 to demonstrate that the lower C1 selectivity attained by Na-Ru/SiO₂ was not due to a difference in conversion.

Author reply: We gratefully appreciate for your professional comment. We have added CO conversion results in Figure 1 and Supplementary Fig. 6. To better illustrate this issue, we compared the product selectivity of Ru/SiO₂ and Na-Ru/SiO₂ catalysts under similar high CO conversion level (*Supplementary Fig. 5*). It was found that a lower C1 selectivity (7.4%) was still maintained even at high CO conversion for Na-Ru/SiO₂, demonstrating that the low C1 selectivity attained by Na-Ru/SiO₂ was not due to a difference in conversion.

In the revised manuscript and supplementary information, Fig. 1C, 1D and S6 were replaced by revised figures. And Supplementary Fig. 5 was added in the revised supplementary information. The following sentences were also added into the revised manuscript:

“Furthermore, the product selectivity was compared at similar conversion levels, as shown in **Supplementary Fig. 5**. Under similar CO conversion of ~70%, the sample of Na-Ru/SiO₂ still exhibited high olefins selectivity of ~76% with suppressed C1 by-products, while a large amount of paraffins with selectivity of ~76% were produced over Ru/SiO₂ case.”

Fig. 1. Catalytic performance for direct syngas conversion to olefins. (C) Product selectivity, CO conversion and olefins yield at different H₂/CO ratios in syngas over

Na-Ru/SiO₂ catalyst at 533 K, 3000 mL·g_{cat.}⁻¹·h⁻¹, and 1.0 MPa. **(D)** Product selectivity, CO conversion and olefins yield at different space velocities over Na-Ru/SiO₂ catalyst at 533 K, H₂/CO ratio of 2 and 1.0 MPa.

Supplementary Fig. 6 | Catalytic performance of Ru/SiO₂ (A) and Na-Ru/SiO₂ (B) at various reaction temperatures.

Supplementary Fig. 5 | Comparison of catalytic performance at similar CO conversion over Ru/SiO₂ and Na-Ru/SiO₂ catalysts. Reaction conditions: 533 K, 3000 mL·g_{cat.}⁻¹·h⁻¹ (Ru/SiO₂), and 1500 mL·g_{cat.}⁻¹·h⁻¹ (0.5Na-Ru/SiO₂), 1 MPa, H₂/CO ratio of 2.

5. Line 178-179 ‘Specially, the homogenous distribution of Na over the catalyst surface may benefit the strong electronic interaction between Ru NPs and Na.’ This is doubtful, because one would think that Na has to be in contact with the Ru NPs (i.e. Figure S10) to promote the Ru active sites and the Na species on the support acted as spectators. From the various characterization data, could the authors (semi)quantify the amount of Na on the support vs. on Ru NPs? This would clarify the role of Na when it is on the support vs. Ru NPs. The authors are further suggested to include HAADF-STEM images and EDX elemental mapping of the spent Ru/SiO₂ and 2Na-Ru/SiO₂ catalysts so as to check for the sensitivity of the Na signal and to prove the above sentence. The experimental loading of Na measured by ICP-OES should also be included.

Author reply: We gratefully appreciate for your professional comment. According to the suggestion of reviewer, HAADF-STEM images and EDX elemental mapping of the spent Ru/SiO₂ and 2Na-Ru/SiO₂ catalysts were included into the *Supplementary Fig. 11*. The loading of Na was also measured by ICP-OES and the result was added as *Supplementary Table 7*.

Based on various characterization, the introduction of Na greatly improved the dispersion of Ru NPs. Specially, the Na promoter was homogeneously distributed on both the SiO₂ support and Ru NPs. In addition, a higher density of Na promoter can be clearly identified as increasing Na loading amount. We agreed with the reviewer that the electronic interaction could be strengthened by increasing the close contact of Na and NPs. However, it is still difficult to tailor the location of Na and it is a great challenge to quantify the amount of Na on the support or on Ru NPs. In addition, sodium migration may happen under FTO working conditions. Whereas, a high Na density for Na-Ru/SiO₂ with increasing Na loading and its homogeneous distribution increased the chance of close contact of Na and Ru NPs.

In the revised manuscript and supplementary information, we have added the HAADF-STEM images and EDX elemental mapping for the spent Ru/SiO₂ and 2Na-Ru/SiO₂ in *Supplementary Fig. 11*. The ICP-OES results were also added in the *Supplementary Table 7*.

Supplementary Fig. 11 | HAADF-STEM images and EDX elemental mapping of various stage catalyst. (A) reduced Na-Ru/SiO₂, (B) spent Na-Ru/SiO₂, (C) spent Ru/SiO₂ and (D) spent 2Na-Ru/SiO₂ catalyst. Ru (green), Na (red), Si (orange).

Supplementary Table 7 | CO chemisorption and ICP results for different Ru-based catalysts after reduction.

Sample	Ru loading ^a (wt.%)	Na loading ^a (wt.%)	Na/Ru molar ratio	d _{XRD} ^b (nm)	d _{TEM} ^c (nm)	D _{TEM} ^d (%)	CO uptake (μmol•g ⁻¹)	Metallic Surface Area (m ² •g _{Ru} ⁻¹)	D _{CO} ^e (%)
Ru/SiO ₂	4.57	0.06	0.06	7.8	7.9	14.2	24.0	23.8	5.3
Na-Ru/SiO ₂	4.18	0.57	0.60	5.3	4.7	23.8	45.5	49.3	11.0
Na-5%Ru(P)/SiO ₂	4.11	0.55	0.58	6.2	4.4	25.5	51.4	56.7	12.7
Na-2Ru(P)/SiO ₂	1.79	0.21	0.52	6.2	5.4	20.7	16.1	40.8	9.1

^a Ru loading and Na loading measured by ICP.

^b Ru⁰ crystallites size calculated by Scherer Formula from XRD.

^c Ru⁰ mean particle size counted by TEM profiles.

^d $D_{\text{TEM}}=1.12/d_{\text{TEM}}$.

^e Dispersion of Ru⁰ nanoparticles calculated by CO chemisorption experiment.

In the revised manuscript, the following sentences were added.

“Furthermore, we found that the introduction of Na greatly improved the dispersion of Ru NPs. Specially, the Na promoter was homogeneously distributed on both SiO₂ support and Ru NPs. A higher density of Na promoter can also be clearly identified as Na loading amount increases (**Supplementary Fig. 11**).”

6. Line 198-199 ‘Based on the linear characteristic of ASF distribution, we can infer that the surface carbide mechanism ...’ This is not so accurate, because the linearity of the ASF distribution is a characteristic of the FTS and FTS technologies, regardless of surface carbide/bulk carbide/CO insertion mechanisms etc. The authors could perhaps refer to computational studies on Ru-based FTS catalysts by the group of Hensen (10.1039/c4cy00483c and 10.1002/anie.201406521) to strengthen their mechanism discussion.

Author reply: We gratefully appreciate for your professional comment. We agree with the review that the linear characteristic of ASF distribution can be obtained over surface carbide/bulk carbide/CO insertion mechanisms. According to the suggestion of reviewer, we have thoroughly read the references of the group of de Jong (*Science*, 2012, 335,835) and the group of Hensen (*Catal. Sci. Technol.*, 2014,4, 3129; *Angew. Chem. Int. Ed.* 2014, 53, 12746). Overall, FTS reaction mechanism is rather complex, and many uncertainties exist on the nature of the reaction intermediates. In the future work, comprehensive density functional theory study and advanced in-situ and time-resolution characterization techniques should be combined together to reveal the nature of the reaction intermediates and reaction mechanism. Based on the DFT study, the group of Hensen pointed out that the surface carbide mechanism explains the formation of long chains hydrocarbon on the stepped Ru surface, while methane would be the

primary hydrocarbon within the CO insertion mechanism. The lower methane selectivity and higher chain-growth-probability indicated that the surface carbide mechanism may be suitable for the Ru-based catalysts in this work.

To avoid the misunderstanding, the following sentences were revised as follow:

“Based on the linear characteristic of ASF distribution and the higher chain-growth probability (α) as well as the ultralow CH_4 selectivity, it can be inferred that the both Ru/SiO₂ and Na-Ru/SiO₂ might follow the analogous reaction mechanism²⁹⁻³¹. This can be rationalized from the simplified surface carbide mechanism (**Supplementary scheme 1**), which is widely accepted for the FTS³². Typically, the dissociated CO would be hydrogenated to form CH_x as the main surface intermediate for chain propagation on metallic Ru surface. The carbon chain grows by coupling of CH_x units to the adsorbed alkyl-chain species. The chain growth is terminated by hydrogenation to produce paraffins or β -hydride abstraction to form olefins.”

In addition, *Supplementary scheme 1* was replaced in the revised supplementary information.

Supplementary scheme 1 | Fischer-Tropsch synthesis based on carbide mechanism³⁵⁻³⁷. The chain growth is terminated by β -hydride elimination or hydrogenation.

7. Line 265-268 regarding the discussion on the WGS activity of Na-Ru/SiO₂. The authors failed to acknowledge that metallic Ru-based FT catalysts, similar to the metallic Co-based catalysts (in ref. 14), have negligible WGS activity at low/moderate CO conversion levels. In Figure 1f, Ru/SiO₂ had no WGS activity and the WGS activity actually increased with increasing Na loading. The authors should clarify that the WGS activity increased with Na promotion and provide a possible explanation on why Na promotion increased WGS activity and what are the possible implications.

Author reply: We gratefully appreciate for your professional comment. We agree with the reviewer that the metallic Ru/SiO₂ catalyst, similar to the metallic Co-based catalysts, have negligible WGS activity. However, the addition of alkali metal promoter, i.e., Na, to the Ru/SiO₂ catalyst slightly increased the CO₂ selectivity. Such phenomenon is commonly observed over metallic Co-based catalysts, where the alkali metal doping also slightly increased CO₂ selectivity (*Ind. Eng. Chem. Res.* 2004, 43, 2391-2398; *Catal. Today*, 2013, 215, 60; *Appl. Catal. B: Environ.*, 2018, 230, 203).

In the revised manuscript, we have clarified that the CO₂ selectivity increased with Na promotion and the possible explanation was also added. In addition, the nature role of Na on CO₂ selectivity deserves a continuous and in-depth studies, and we will clarify it in the next research work by combining in-situ characterization techniques and DFT calculation.

In the revised manuscript, the following sentences and figure were revised and added:

“In view of the ultralow intrinsic WGS reactivity of metallic Ru, we inferred that the Na-promoted Ru catalyst with metallic Ru as active phase possessed similar property. To verify this viewpoint, a WGS reaction probe experiment was performed as shown in **Supplementary Fig. 21.”**

“Furthermore, the Na doping also slightly increased CO₂ selectivity for Ru/SiO₂ catalyst (Fig. 1F**). Prior studies have revealed that the H-assisted CO_{ad} dissociation route prevails on Ru cluster surface with near-saturation CO* coverage during FTS**

process²⁶, which features the preferential formation of H₂O instead of CO₂ as the primary oxygen removing pathway (**Supplementary Scheme 2**), similar to those observed over metallic Co-based catalysts¹⁵. **The suppressed reactivity of chemisorbed H₂ and increased CO adsorption may slightly promote the generation of CO₂ after introducing Na to the Ru/SiO₂ catalyst.”**

8. Line 471-472 regarding the scale-up operation. In the microreactor, plug-flow conditions appear to be fulfilled (reactor inner diameter >10 times catalyst particle sieve fraction, catalyst bed height >50 times catalyst particle sieve fraction). However, this does not appear to be the case for the pilot-scale reactor as the reactor inner diameter was only 4 to 6 times the dimensions of the extrudate. To demonstrate the success of the scale-up operation to claim ‘industry-relevant testing’, the pilot-scale reactor should also be operated under plug flow conditions and a direct comparison of the Na-2%Ru(P)/SiO₂ catalyst performance in the microreactor and the pilot-scale reactor would be appreciated.

Author reply: We gratefully appreciate for your valuable comment. According to the suggestion of reviewer, in order to fulfill the requirements of plug-flow conditions (reactor inner diameter >10 times catalyst particle sieve fraction, catalyst bed height >50 times catalyst particle sieve fraction), the evaluation test in pilot-scale fixed-bed reactor was repeated with smaller particle size. Na-2%Ru(P)/SiO₂ catalyst with particle size of 12 – 20 mesh (0.85 – 1.7 mm) was loaded into a pilot-scale fixed-bed reactor (internal diameter: 19 mm; length: 1180 mm) for FTO reaction. The identical catalyst was also evaluated in the microreactor.

Under the reaction conditions of 538 K, 1.0 MPa, 3000 mL·g_{cat.}⁻¹·h⁻¹, H₂/CO ratio of 2, the Na-2%Ru(P)/SiO₂ catalyst showed 47.7% of CO conversion with 73.2% of olefins selectivity and < 5% of C1 by-products in a microreactor. When being evaluated in a pilot-scale reactor, the same catalyst exhibited similar catalytic performance, which showed 40.5% of CO conversion with 72.5% of olefins selectivity. The total selectivity of C1 by products including CH₄ and CO₂ was still less than 5%. This result suggests that the Na-2%Ru(P)/SiO₂ catalyst shows a promising industrial application.

In the revised manuscript, the relevant parameters were revised in the industry-relevant testing of *Methods* section. And the corresponding catalytic data and Figure were replaced and the following sentences were added.

Supplementary Fig. 23 | Comparison of catalytic results evaluated in a pilot-scale reactor and microreactor. (A) CO conversion and product selectivity. (B) Chain-growth probability (α) and (C, D) hydrocarbons distribution of the Na-2%Ru(P)/SiO₂ catalyst in the pilot-scale reactor (C) and microreactor (D). Reaction conditions: 538 K, 1.0 MPa, 3000 mL·g_{cat.}⁻¹·h⁻¹ and H₂/CO ratio of 2.

“Supplementary Notes:

The Na-2%Ru(P)/SiO₂ catalyst was evaluated in a pilot-scale reactor (12 – 20 mesh) and microreactor (40 – 60 mesh), respectively, under reaction conditions of 538 K, 1.0 MPa, 3000 mL·g_{cat.}⁻¹·h⁻¹, H₂/CO ratio of 2. As shown in Supplementary Fig. 23, the olefins selectivity in total products reached up to 72.5% while the sum selectivity of undesired CH₄ and CO₂ was suppressed within 5% at CO conversion of 40.5% and TOF of 0.312 s⁻¹ in the pilot-scale reactor, which is very similar to that in microreactor. The CH₄ selectivity for both reactors was much lower than the value predicted by the

classic ASF model. Moreover, a chain-growth probability at around 0.76 was obtained in both pilot-scale reactor and microreactor, demonstrating the as-obtained catalyst is very suitable to produce long-chain olefins. In addition, a similar hydrocarbon distribution was also obtained, confirming that the pellet Na-2%Ru(P)/SiO₂ catalyst shows a promising industrial application with high olefins yield and low fraction of undesired C1 by-products.”

9. Spelling mistakes: line 78 (‘predicated’ = predicted), line 250 (‘specular’ = speculated), line 421 (‘stand’ = standard).

Author reply: We gratefully appreciate for your kind reminder. We have corrected them in the revised manuscript.

10. Experimental methodology for x-ray spectroscopies is missing?

Author reply: We gratefully appreciate for your kind reminder. The experimental methodology for x-ray spectroscopies was added in the revised manuscript as following.

“X-ray absorption fine structure (XAFS) data was performed at the BL14W1 of Shanghai Synchrotron Radiation Facility (SSRF), China. The storage ring of the SSRF was operated at 3.5 GeV with a maximum current of 230 mA. All data was acquired at the Ru K-edge in transmission mode. X-ray absorption near-edge spectroscopy (XANES) and extended X-ray fine-structure (EXAFS) spectroscopy of samples were collected under ambient condition using a fixed-exit double-crystal Si (111) monochromator. Catalyst sample was pressed into pellets within LiF, and then placed inside a stainless steel in situ cell which was surrounded by a heater. Reduction of the Na-Ru/SiO₂ catalyst was carried out by heating the in-situ cell at 10 K/min in the following pure H₂ up to 573 K, during which the XAFS spectra were measured at 298 K, 423 K and 573 K, respectively. Then the reduced Na-Ru/SiO₂ catalyst was treated by syngas (H₂/CO=2) in the in-situ cell at 533 K for 30 min and the XAFS spectra was collected. The data analysis was performed using IFEFFIT software package according to standard data analysis procedures³⁴. The energy was calibrated by collecting spectra

of Ru foil standard sample. After appropriate background subtraction, the k^2 -weighted EXAFS spectra of the Ru K-edge data ranges were assessed based on the quality of data generally between $k = 3 - 12 \text{ \AA}^{-1}$ and for $R = 1 - 3 \text{ \AA}$. All data fitting was performed by Artemis program in IFEFFIT. The value of the passive electron amplitude reduction factor, S_0^2 , was determined to be 0.75 for Ru, by a fit of a reference Ru foil with a fixed coordination number of 12 to reflect the HCP structure of Ru.”

REVIEWERS' COMMENTS

Reviewer #1 (Remarks to the Author):

This manuscript has been significantly improved based on the comments from the reviewers. I would recommend the publication in this journal.

Reviewer #2 (Remarks to the Author):

The authors have properly addressed the comments raised by this (and the other) reviewer(s) and introduced the required changes in the main manuscript and supporting information. This has even further improved the already high scientific value of the work. From my side, the present revised version of the manuscript is suitable to be published in Nature Commun.

Reviewer #3 (Remarks to the Author):

The reviewer appreciates the efforts and thoughts that the authors put in to the revision, and congratulates the authors on this excellent piece of work. All reviewers' feedback have been addressed by the authors so this revised version is recommended for publication in Nature Communications.

Jingxiu Xie

Point-by-point responses to all the comments from referees

Reviewer #1

Comments:

This manuscript has been significantly improved based on the comments from the reviewers. I would recommend the publication in this journal.

Author reply:

Thank you very much for reviewing our manuscript, and providing very helpful suggestions for us to improve the quality of this manuscript.

Reviewer #2

Comments:

The authors have properly addressed the comments raised by this (and the other) reviewer(s) and introduced the required changes in the main manuscript and supporting information. This has even further improved the already high scientific value of the work. From my side, the present revised version of the manuscript is suitable to be published in Nature Commun.

Author reply:

Thank you very much for reviewing our manuscript. We appreciate your helpful comments, which benefit us to improve the quality of our work.

Reviewer #3

Comments:

The reviewer appreciates the efforts and thoughts that the authors put in to the revision, and congratulates the authors on this excellent piece of work. All reviewers' feedback have been addressed by the authors so this revised version is recommended for publication in Nature Communications.

Jingxiu Xie

Author reply:

Thank you very much for reviewing our manuscript, and proposing many

constructive comments for us to improve the quality of this work.